

# Mid-Holocene climate at mid-latitudes: assessing the impact of the Saharan greening

Marco Gaetani[1], Gabriele Messori[2,3], Francesco S.R. Pausata[4], Shivangi Tiwari[4], M. Carmen Alvarez Castro[5], Qiong Zhang[6]

[1]University School for Advanced Studies IUSS, Pavia, Italy

[2]Dept. of Earth Sciences and Centre of Natural Hazards and Disaster Science (CNDS), Uppsala University, Sweden

[3]Dept. of Meteorology and Bolin Centre for Climate Research, Stockholm University, Stockholm, Sweden

[4]University of Quebec in Montreal, Canada

[5]Dept. of Physical, Chemical and Natural Systems, Pablo de Olavide University, Seville, Spain.

[6]Department of Physical Geography and Bolin Centre for Climate Research, Stockholm University, Stockholm, Sweden

*Correspondence to*: Marco Gaetani (marco.gaetani@iusspavia.it)

**Abstract.** During the first half of the Holocene (11,000 to 5,000 years ago) the Northern Hemisphere experienced a strengthening of the monsoonal regime, with climate reconstructions robustly suggesting a greening of the Sahara region. Paleoclimate archives also show that this so-called African Humid Period (AHP) was accompanied by changes in the climate

conditions at mid to high latitudes. However, inconsistencies still exist in reconstructions of the mid-Holocene (MH) climate at mid-latitudes, and model simulations provide limited support to reduce these discrepancies. In this paper, a set of simulations performed with a climate model is used to investigate the hitherto unexplored impact of the Saharan greening on mid-latitude atmospheric circulation during the MH. Numerical simulations show a year-round impact of the Saharan greening on the main circulation features in the Northern Hemisphere, especially during boreal summer when the African monsoon develops. Key

findings include a westward shift of the global Walker Circulation, leading to a modification of the North Atlantic jet stream in summer and the North Pacific jet stream in winter. Furthermore, the Saharan greening modifies the atmospheric synoptic circulation over the North Atlantic, transitioning the North Atlantic Oscillation phase from prevailingly positive to neutral-to-negative in winter and summer. This study provides a first constraint on the Saharan greening influence on northern mid-latitudes, indicating new opportunities for understanding the MH climate anomalies in regions such as North America and

Eurasia.





## 1 Introduction

The early and middle Holocene (11,000 to 5,000 years ago) were characterised by the summer solstice occurring close to the perihelion of the Earth's orbit, which led to increased insolation during boreal summer and consequent modifications in climate seasonality. This period is often referred to as the "Holocene thermal optimum" that resulted in remarkable climate and environmental changes in the Tropics and at mid and high latitudes. The Northern Hemisphere experienced a reinforcement of the global monsoonal regime (Jiang et al., 2015; Wu and Tsai, 2021; Bosmans et al., 2012; Zhao and Harrison, 2012; Haug et al., 2001; Yuan et al., 2004; Wang et al., 2008). This monsoonal intensification particularly manifested in Africa, leading to the so-called "African Humid Period" (AHP) and the subsequent greening of the Sahara (Pausata et al., 2020; Claussen et al., 2017, 1999; Claussen and Gayler, 1997; Adkins et al., 2006; Tierney and DeMenocal, 2013; Tierney et al., 2017, 2011; Hoelzmann et al., 1998; Larrasoaña et al., 2013).

At mid-latitudes, paleoclimate proxies suggest a complex climatic evolution: gradual cooling of the northeast Atlantic, contrasted with a warming in the western subtropical Atlantic, the eastern Mediterranean and the northern Red Sea from the early to the middle Holocene (Andersson et al., 2010; Rimbu et al., 2003). These changes were accompanied by prevailing negative phases of the Arctic and North Atlantic Oscillations (AO and NAO, respectively) (Rimbu et al., 2003; Olsen et al., 2012; Nesje et al., 2001). The proxy records further indicate region-specific climatic deviations from the pre-industrial climate: eastern North America and Scandinavia likely experienced warmer and drier; Western Europe, colder winters and warmer summers; Central Europe, an overall warming; the Mediterranean, colder and rainier conditions; and central Asia, increased annual rainfall, warmer winters and colder summers **(Fig. 1)** (see e.g., Cronin et al., 2005; Bartlein et al., 2011; Scholz et al., 2012; Samartin et al., 2017; Davis et al., 2003). However, the interpretation of these climatic changes, particularly on temperature and precipitation patterns, as indicated by proxies, seem potentially inconsistent with the suggested changes in the atmospheric circulation (e.g., the positive-to-negative shift in the NAO/AO phase). Furthermore, differences exist in the estimation of both timing and magnitude of the Holocene thermal maximum at mid to high latitudes (Kaufman et al., 2004; Renssen et al., 2009; Cartapanis et al., 2022).



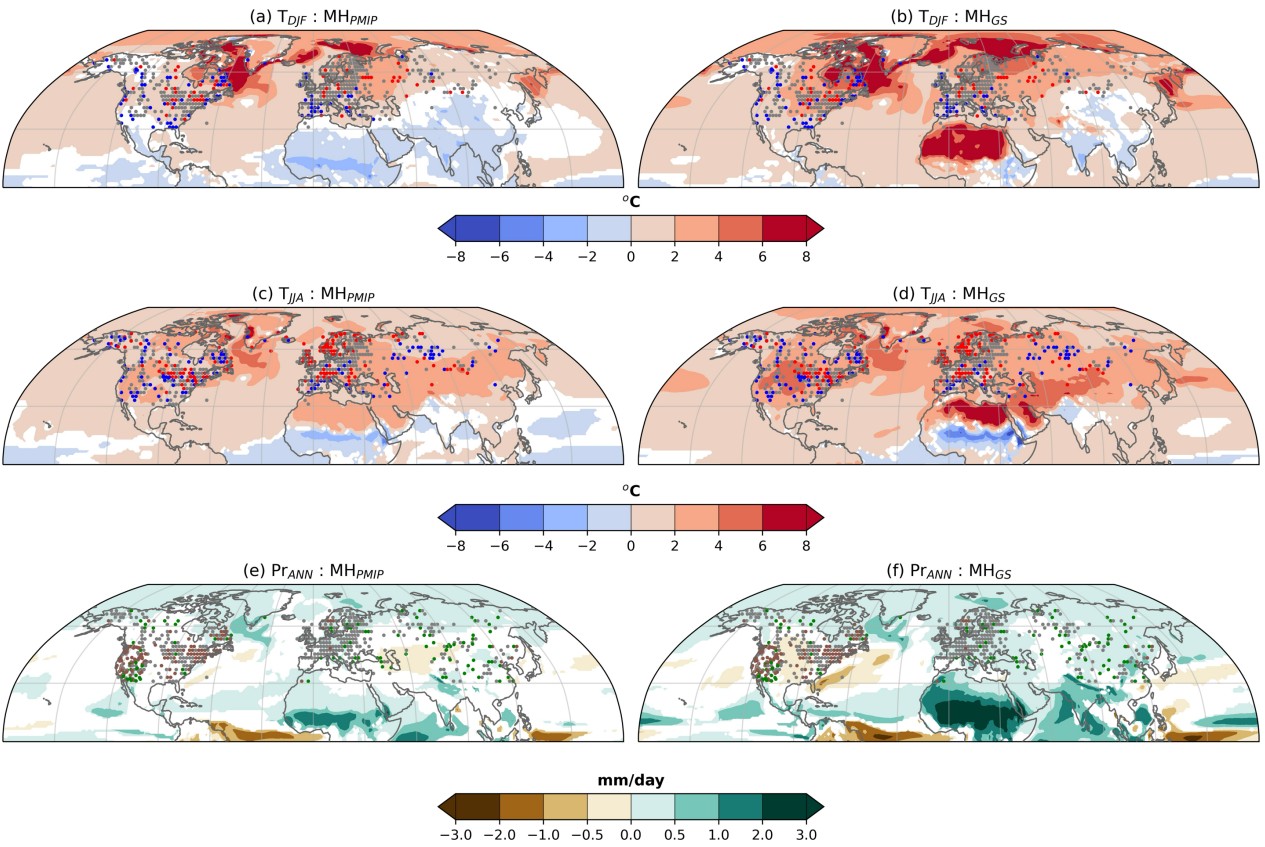

**Figure 1. Model vs. proxy reconstruction comparison: Changes in (a, b) winter temperature, (c, d) summer temperature, and (e, f) annual precipitation between MH$_{PMIP}$ and PI (left) and between MH$_{GS}$ and PI (right). Coloured shading shows anomalies significant at the 95% confidence level after the Student's $t$-test. Filled dots show proxy sites and their MH signature relative to the PI. Red dots indicate warmer signature, blue dots indicate cooler signature, brown dots indicate drier signature, green dots indicate wetter signature, while grey dots indicate no change or inconclusive signature. Model simulations and proxy reconstructions are described in section 2.**

In this context, climate models struggle to constrain the climate conditions associated with the Holocene thermal optimum. In the northern monsoon regions, precipitation increase is generally underestimated, while summer warming at mid to high latitudes is overestimated (Harrison et al., 2014). To explain the limitations of climate models in representing the middle Holocene climate, several studies pointed to the role of the vegetation together with other feedbacks at tropical and higher latitudes in modulating the climate response to the orbital forcing (Pausata et al., 2016; Swann et al., 2012, 2014; Chandan and



Peltier, 2020). In particular, the remarkable greening of the Sahara influenced the regional and global climate during the middle
Holocene. Modelling studies demonstrate that the resulting reduction in albedo and dust emission, along with enhanced water
recycling associated with the increased vegetation cover, were key ingredients in maintaining the intensified African monsoon
regime during the AHP (Messori et al., 2019; Gaetani et al., 2017; Pausata et al., 2016; Tierney et al., 2017), as well as in
reinforcing the global monsoon system, and modifying the tropical cyclone activity and the El Niño/Southern Oscillation
variability (Sun et al., 2019; Pausata et al., 2017b, a; Swann et al., 2014; Piao et al., 2020). However, while paleoclimate
modelling of the Green Sahara mainly focused on the impact in the Tropics and sub-tropics, the studies on climate responses
at mid-latitudes is still limited.

The objective of this paper is to study the impact of the Saharan greening on the Northern Hemispheric mid-latitude
atmospheric circulation and associated climate variability during the middle Holocene (MH). To achieve this, climate model
is used to investigate the relevant underlying mechanisms. Moreover, the model-proxy agreement at mid-latitudes when
considering vegetated Sahara with reduced dust emission is evaluated. This study focuses on the analysis of the winter
(December to February, DJF) and summer (June to August, JJA) seasons in the Northern Hemisphere.

## 2 Data and methods

In this paper, the climate experiments described in Pausata et al. (2016) are analysed. These simulations were conducted using
version 3.1 of the atmosphere–ocean fully coupled climate model EC-Earth (Hazeleger et al., 2010). The atmospheric model
is based on the Integrated Forecast System (IFS cycle 36r4) (https://www.ecmwf.int/en/forecasts/documentation-and-support),
including the H-TESSEL land model (van den Hurk et al., 2000). The simulations are run at T159 horizontal spectral resolution
(~1.125°, approximately 125 km) with 62 vertical levels. The atmospheric component is coupled by the OASIS 3 coupler
(Valcke 2006) to the Nucleus for European Modeling of the Ocean (NEMO) version 2 (Madec 2008), and the Louvain-la-
Neuve Sea Ice Model version 3 (LIM3) (Vancoppenolle et al., 2009). The ocean component NEMO has a nominal horizontal
resolution of 1° and 46 vertical levels.

A 700-year pre-industrial (PI) control simulation following the CMIP5 protocol (Taylor et al., 2012) was conducted to provide
initial conditions to the MH simulations, which were run for about 300 years (climate equilibrium is reached after 100–200
years, depending on the experiment). For each experiment, data from the last 30 years are retained for analysis. The
atmospheric dust concentration representative of the PI conditions is prescribed in the PI simulation by using the long-term
monthly mean (1980-2015) from the MERRAero product. This dataset includes the radiative coupling of the GEOS-5 climate
model to the GOCART aerosol module and assimilates satellite retrievals of aerosol optical depth (AOD) from the MODIS
sensor. Details on the MERRAero dataset are available at https://gmao.gsfc.nasa.gov/reanalysis/merra/MERRAero/. In the
analysed simulations, the EC-Earth3.1 simulates the direct effect of dust on the atmospheric radiative balance, though it does
not simulate indirect effect on cloud formation and microphysics.





A MH simulation is run following the PMIP3 protocol (MH$_{PMIP}$): the orbital forcing is set to MH values (6,000 years BP); the solar constant, land cover, ice sheets, topography, and coastlines are set to PI conditions, as well as the greenhouse gas concentrations, with the exception of the methane concentration set at 650 ppb. An additional MH simulation is run with prescribing Green Sahara (GS) conditions, namely vegetated surface and reduced dust emission in the Sahara–Sahel region (MH$_{GS}$) (11°–33°N, 15W°– 35°E). Land cover in the Sahara is prescribed to be evergreen shrub with a leaf area index (LAI) of 2.6. This vegetation cover leads to an average decrease in surface albedo from 0.30 during the PI period to 0.15. Surface roughness and soil wetness are prescribed to PI values. Dust emission typical of the AHP is simulated by prescribing an 80% reduction in dust concentration throughout the troposphere (up to 150 hPa) over a broad area around the Sahara region [see Fig. S1 in Gaetani et al. (2017)]. Although this experimental design is highly idealized, it is firmly based on paleoclimatic reconstructions for the MH from both dust (deMenocal *et al.*, 2000; McGee *et al.*, 2013) and pollen archives (Lézine *et al.*, 2011; Hély *et al.*, 2014). The experimental setup is summarized in **Table 1**.

**Table 1. Experimental set-up. Vegetation type, surface albedo and LAI refer to the Sahara region.**

| Simulation | Orbital forcing | GHGs | Vegetation type | Albedo | LAI | Dust concentration |
|---|---|---|---|---|---|---|
| **PI** | Present day | PI | Desert | 0.30 | 0 | 1980-2015 climatology |
| **MH$_{PMIP}$** | 6ka | MH | Desert | 0.30 | 0 | 1980-2015 climatology |
| **MH$_{GS}$** | 6ka | MH | Evergreen shrub | 0.15 | 2.6 | 80% reduced |

The climates in the MH$_{PMIP}$ and MH$_{GS}$ simulations are compared with multi-archival proxy reconstructions of continental seasonal temperatures and annual precipitation. To this end, statistically significant anomalies (MH minus PI) simulated by the model are compared with MH signatures as indicated by a compilation of proxy records. This compilation not only builds upon datasets previously available from Bartlein et al. (2011) and Hermann et al. (2018) but also enriches it with 89 additional multi-archival, multi-proxy records, as listed in **Table A1**. To assess the agreement between proxy data and model simulations, the MH signatures at each proxy site are assigned to categories such as wetter/drier/no change or inconclusive and warmer/cooler/no change or inconclusive. Proxy-model agreement is quantified using the Cohen's Kappa index, following DiNezio and Tierney (2013). The Cohen's Kappa index is calculated separately over four regions: North America, Pacific Coast (180–100°W; 30–70°N); North America, Atlantic Coast (100–30°W, 30–70°N); Europe and Mediterranean (20°W–50 °E, 30–70°N); and Asia (50–180°E, 30–70°N) (Fig. A1).

Part of the analysis of atmospheric circulation variability focuses on the North Atlantic region, as in the present climate this relates to climate variability in both Eurasia and North America. The main modes of variability of the North Atlantic





atmospheric circulation are extracted at the monthly and daily time scales, by applying a principal component analysis (PCA) (Wilks, 2019) to the geopotential height anomalies at 500 hPa over the domain (80W°– 30°E, 20°–80°N). In order to assess the modifications in the circulation patterns in the MH experiments relative to the PI simulation, all geopotential height anomalies are computed with respect to the PI climatology. The PCA is then applied to concatenated anomaly data from all

three simulations. That is, the data matrix, $Z$, used to compute the covariance matrix, which is then used to find eigenvectors

and eigenvalues (Wilks, 2019), is defined as: $Z = \begin{bmatrix} PI \\ MH_{PMIP} \\ MH_{GS} \end{bmatrix}$. Here $PI$, $MH_{PMIP}$ and $MH_{GS}$ are themselves matrices, with

longitude and latitude data (55×30 grid points) arranged in columns and the time steps (30×3 or 30×92 for the monthly and daily analyses, respectively) arranged in rows.

The first EOF and the associated expansion coefficient time series derived from the PCA of the monthly anomalies are used

to represent, respectively, the spatial pattern of the North Atlantic Oscillation (NAO) and its time evolution (NAO index, NAOI) as the NAO is acknowledged as the dominant mode of the atmospheric circulation in the North Atlantic region (Hurrell et al., 2003). At daily time scale, the first several PCA modes (first 7 modes in winter and first 8 modes in summer, accounting for 70% of the circulation variability) are used to classify the weather variability affecting North America, the North Atlantic and Europe. Because the selected EOFs show poor separation at the daily time scale, a rotation is applied (Wilks, 2019), and

the rotated EOFs are used with a k-means classification algorithm (Wilks, 2019) to identify the four canonical weather regimes (WRs) characterising the synoptic atmospheric variability in the North Atlantic (Michelangeli *et al.*, 1995).

In order to facilitate the proxy-model comparison, all climate simulations are remapped to the 2-degree grid of the Bartlein et al. (2011) dataset, which is the only proxy dataset on a regular grid used in this study.

**3 Results**

**3.1 Climate response to Saharan greening in the Northern Hemisphere**

In this section, the DJF and JJA climatological responses of near-surface temperature, precipitation and atmospheric circulation to changes in both the MH orbital parameters alone, and in combination with prescribed Saharan greening, are shown. These responses are quantified by comparing the differences between the MH_PMIP and MH_GS experiments against the PI simulation,

respectively.

As a consequence of the change in the orbital parameters, the Northern Hemisphere displays a significantly warmer 2-m temperature in both seasons and experiments **(Fig. 1a-d and Fig. 2)**. The warming is significantly more pronounced when Saharan greening is prescribed, in both winter and summer **(Fig. 1a-d and Fig. 2)**. In winter, the warming peaks in the Arctic region, presumably due to the loss of sea ice **(Fig. 1a, b)**. Moreover, in the MH_GS experiment, the reduced albedo associated

with the vegetation cover increases the radiative forcing in the Saharan region, resulting in a warming effect in the northern tropics **(Fig. 1a, b and Fig. 2a)**. In summer, the Northern Hemisphere is homogeneously warmer from the polar regions to the





subtropics **(Fig. 1c, d and Fig. 2b)**. The surface cooling associated with the intensification of the African monsoon is visible in Northern Africa in both the experiments and is more pronounced in the MH$_{GS}$ experiment **(Fig. 1c, d)** [see also Pausata et al. (2016); Gaetani et al. (2017)].


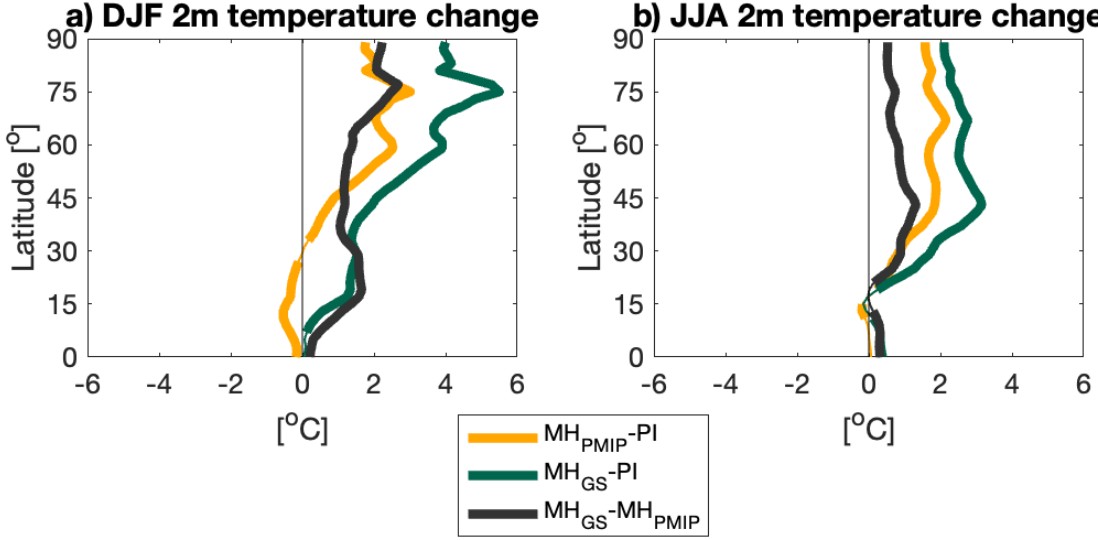

**Figure 2. Climatological latitudinal mean changes in 2-m temperature in the MH$_{PMIP}$ (yellow lines) and MH$_{GS}$ (green lines) simulations relative to the PI simulation, and in the MH$_{GS}$ simulation relative to the MH$_{PMIP}$ simulation (black lines), in boreal (a) winter and (b) summer. Thicker lines display anomalies significant at the 95% confidence level after a Student's *t* test.**


Precipitation in the Northern Hemisphere mid to high latitudes shows a significant increase in winter and summer, in both the MH experiments, with a significant intensification when the Saharan greening is prescribed **(Fig. 1e, f and Fig. 3)**. This precipitation response is associated with a slowing down of the westerly upper tropospheric flow in the subtropics, along with a reinforcement at mid-latitudes **(Fig. A2)**. The precipitation is also significantly enhanced in the northern Tropics for both the MH experiments, with a vegetated Sahara again resulting in a further significant increase relative to MH$_{PMIP}$ **(Fig. 1e, f and Fig. 3)**. In particular, both MH experiments show an intensification of the boreal summer monsoonal regime, which is accompanied by a northward shift of the precipitation belt **(Fig. 1e, f and Fig. 3)**.





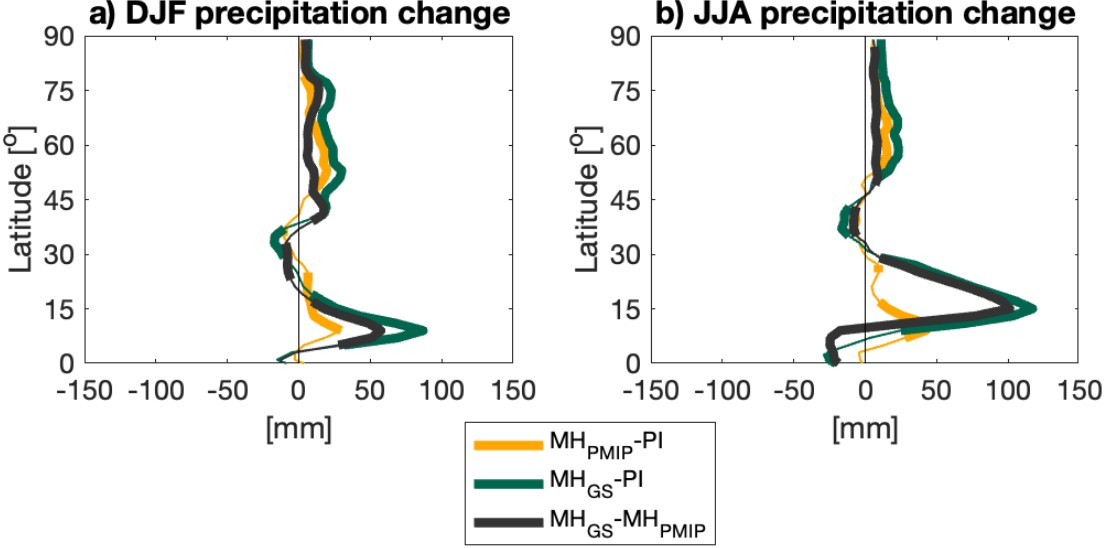

**Figure 3. Climatological latitudinal mean changes in precipitation in the MH$_{PMIP}$ (yellow lines) and MH$_{GS}$ (green lines) simulations with respect to the PI simulation, and in the MH$_{GS}$ simulation with respect to the MH$_{PMIP}$ simulation (black lines), in boreal (a) winter and (b) summer. Thicker lines display anomalies significant at the 95% confidence level after a Student's *t* test.**

The winter precipitation response in the MH experiments is characterised by significant dry anomalies in central Asia and significant wet anomalies at high latitudes in North America and Eurasia, which are scattered in the MH$_{PMIP}$ simulation and widespread in the MH$_{GS}$ experiment **(Fig. A2a, b)**. The presence of vegetated Sahara also compensates the dry anomalies simulated in northern tropical Africa in the MH$_{PMIP}$ experiment **(Fig. A2a, b)**. The meridional profile of the westerly upper tropospheric flow to the west of the North American continent (corresponding to the end of the North Pacific storm track) shows an intensification and a zonalisation of the jet stream in both MH simulations with respect to the PI experiment **(Fig. 4a)**, suggesting a modification in the location and magnitude of the storm track affecting the North American west coast. In the North Atlantic, to the west of the Eurasian continent, both MH simulations show a slight weakening of the westerly wind at the mid-latitudes, accompanied by a slight intensification at the sub-polar latitudes **(Fig. 4b)**, suggesting a possible modification of the circulation pattern over the North Atlantic. However, the prescribed vegetation in the Sahara does not significantly influence the winter dynamics of the jet stream beyond the changes induced by the orbital parameters alone. At global scale, the dynamical signature of the MH experiments in winter is marked by a westward shift of the Walker circulation, as shown by the changes in the velocity potential and divergent wind in the upper troposphere **(Fig. 5a, b)**.

In summer, both MH experiments show wet anomalies in the monsoonal region, as well as the tropical North Atlantic and equatorial Pacific **(Fig. A2c, d)**. The response is stronger in the MH$_{GS}$ simulation due to the effect of Saharan greening on the



African monsoon **(Fig. A2c, d)**. Moreover, in the MH$_{GS}$ experiment, a significant drying at subtropical latitudes is simulated
190 in the North Pacific, North America and the North Atlantic **(Fig. A2d)**. The meridional profile of the westerly upper
tropospheric flow shows significant changes in the MH$_{GS}$ simulation with respect to the PI and MH$_{PMIP}$ experiments.
Specifically, to the west of the North American continent, there is a significant weakening of the mid-latitude westerlies,
accompanied by an intensification at the sub-polar latitudes **(Fig. 4c)**. This suggests a northward shift in the jet stream and a
modification in the location and intensity of the storm track affecting the North American west coast when Saharan greening
195 is prescribed. To the west of the Eurasian continent, there is a significant reinforcement and southward shift of the subpolar jet
stream **(Fig. 4d)**, indicating a shift in the location and magnitude of the North Atlantic storm track. Similar to the winter season,
the global Walker circulation shows a westward shift in summer **(Fig. 5c, d)**. The MH$_{GS}$ experiment shows a stronger response,
which could be expected, but also shows an atmospheric bridge in the upper troposphere, represented by the easterly divergent
wind anomalies. This bridge connects a divergence anomaly in the Indian Ocean with a convergence anomaly in the tropical
200 North Atlantic, as shown by the negative and positive velocity potential anomalies, respectively **(Fig. 5d)**. This feature is
reflected in the simulated merging of the mid-latitude and subtropical jet streams in the North Atlantic **(Fig. 4d and A2d)**,
suggesting a potential mechanism connecting the reinforcement of the monsoonal regime with modifications of the mid-
latitude circulation.

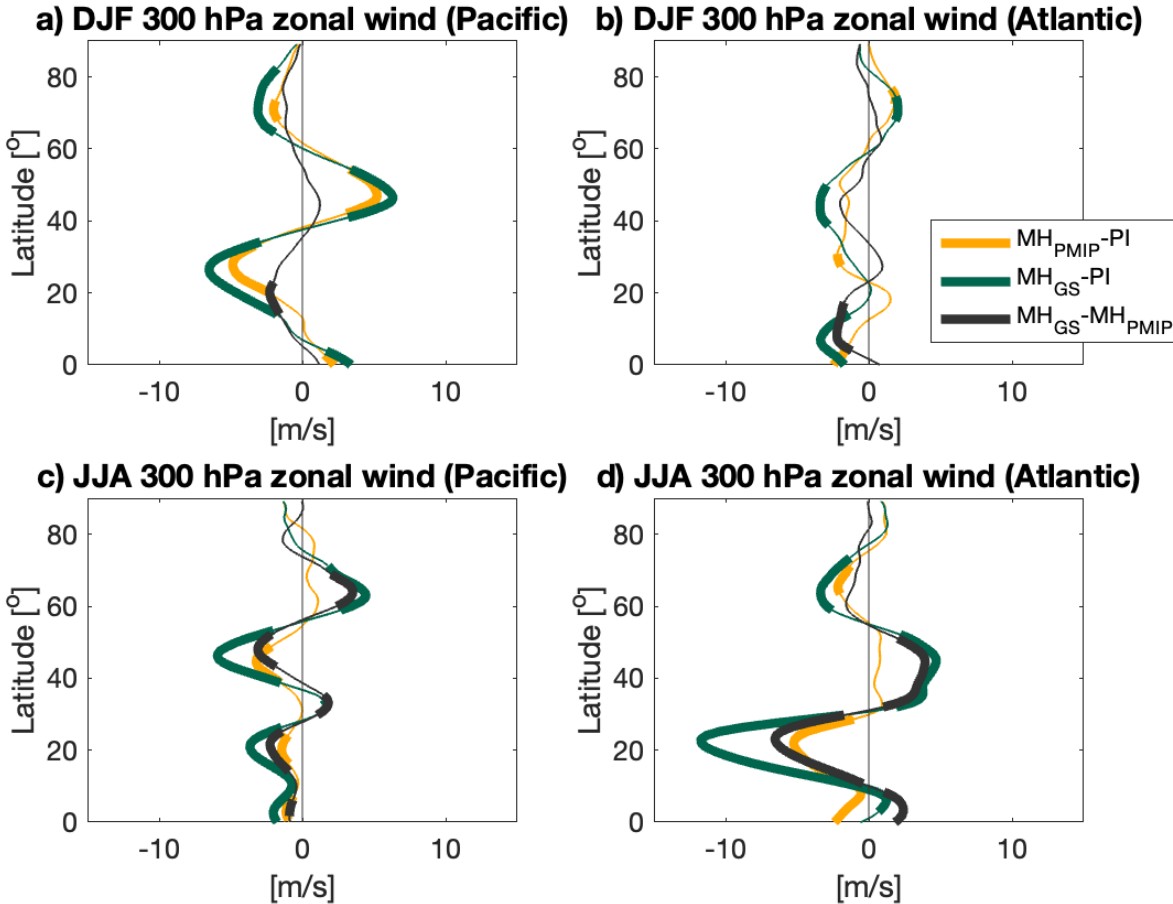

**Figure 4. Climatological latitudinal mean changes in the zonal wind at 300 hPa in the MH$_{PMIP}$ (yellow lines) and MH$_{GS}$ (green lines) simulations with respect to the PI simulation, and in the MH$_{GS}$ simulation with respect to the MH$_{PMIP}$ simulation (black lines), at (a, c) 150°W (North Pacific) and (b, d) 30°W (North Atlantic), in boreal (a, b) winter and (c, d) summer. Thicker lines display anomalies significant at the 95% confidence level after a Student's _t_ test.**



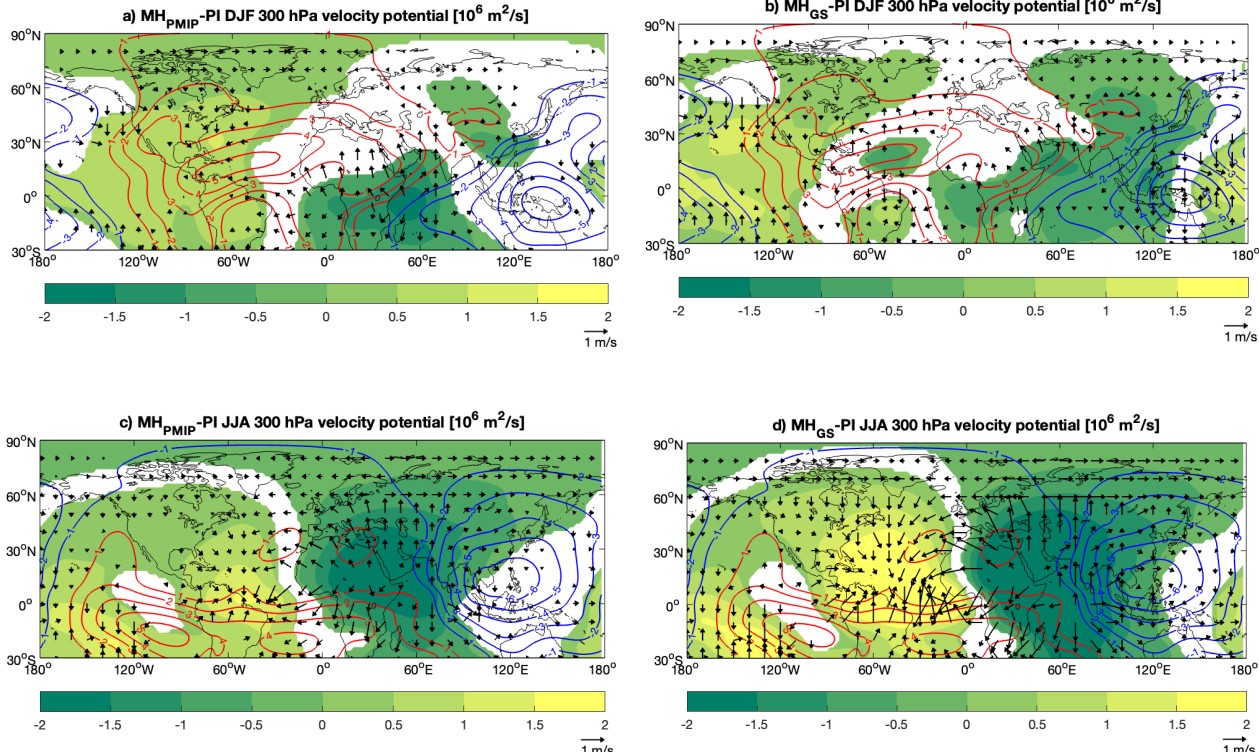

**Figure 5. Changes in the velocity potential (shadings) and divergent wind at 300 hPa (vectors) in the MH$_{PMIP}$ and MH$_{GS}$ simulations in (a, b) DJF and (c, d) JJA with respect to the PI simulation. Red and blue contours display the climatological pattern of velocity potential in the PI simulation. Only locations and vectors showing statistically significant velocity potential and divergent wind anomalies, at the 95% level of confidence after a Student's $t$ test, are shown.**

## 3.2 Changes in the North Atlantic Oscillation

The NAO is the main mode of atmospheric variability influencing climate patterns in the North Atlantic, Europe, and North America (Ambaum et al., 2001; Hurrell et al., 2003; Chartrand and Pausata, 2020). In this section, the changes in NAO variability in the MH experiments with respect to the PI simulation are examined.

In winter, the NAO pattern identified in the model simulations reflects the canonical pattern described in the literature, with a strong meridional geopotential dipole in the North Atlantic explaining 28% of the total variability (Ambaum et al., 2001; Hurrell et al., 2003) **(Fig. A3a)**. In the PI simulation, the NAOI is characterised by a distribution skewed towards positive values **(Fig. 6a)**. The NAO positive phase in the PI simulation is associated with warm anomalies in central and northern Europe and the east coast of North America, and cold anomalies in North Africa **(Fig. 7a)**. Moreover, this phase correlates with dry conditions in southern Europe and wetter conditions in Scandinavia **(Fig. 7b)**. Consistent with previous findings (e.g.,





Nesje et al., 2001; Rimbu et al., 2003; Olsen et al., 2012), changes in orbital parameters and the Saharan greening lead to a shift of the NAOI phase towards negative values in MH$_{GS}$ **(Fig. 6a)**. A Kolmogorov-Smirnov test confirms that these shifts in

the NAOI distributions in both MH$_{PMIP}$ and MH$_{GS}$ experiments are statistically significant when compared to the PI simulation ($p<0.02$). However, the difference in the NAOI distributions between the MH$_{GS}$ and MH$_{PMIP}$ experiments is less significant ($p<0.11$). Circulation and surface anomaly patterns associated with the NAO positive phase in the MH$_{PMIP}$ (not shown) and the MH$_{GS}$ experiments **(Fig. 7c, d)** are very similar. The tendency towards a prevailing neutral-to-negative NAO phase in the MH simulations is then expected to result in colder winters in central and northern Europe and eastern North America, and warmer

conditions in northern Africa, along with wetter southern Europe and drier eastern North America and Scandinavia **(Fig. 7c, d)**. In particular, the thermal and rainfall anomalies are more pronounced when Saharan greening is taken into account, due to the significant difference in the NAO phase shift with respect to the PI period.

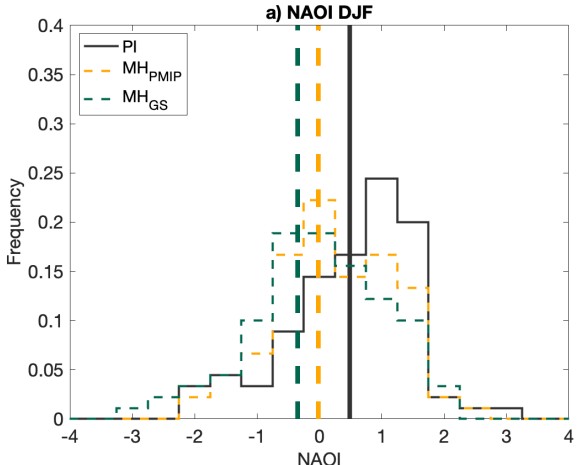
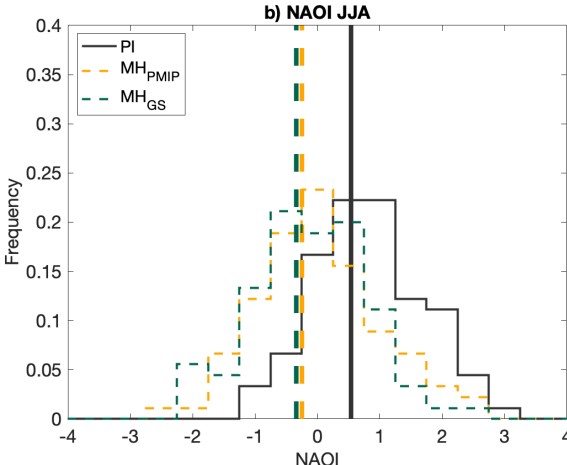

**Figure 6. Distributions of the NAOI in the PI simulation (solid line), and in the MH$_{PMIP}$ and MH$_{GS}$ experiments (dashed lines), in (a) winter and (b) summer; the vertical lines indicate the medians of the distributions.**



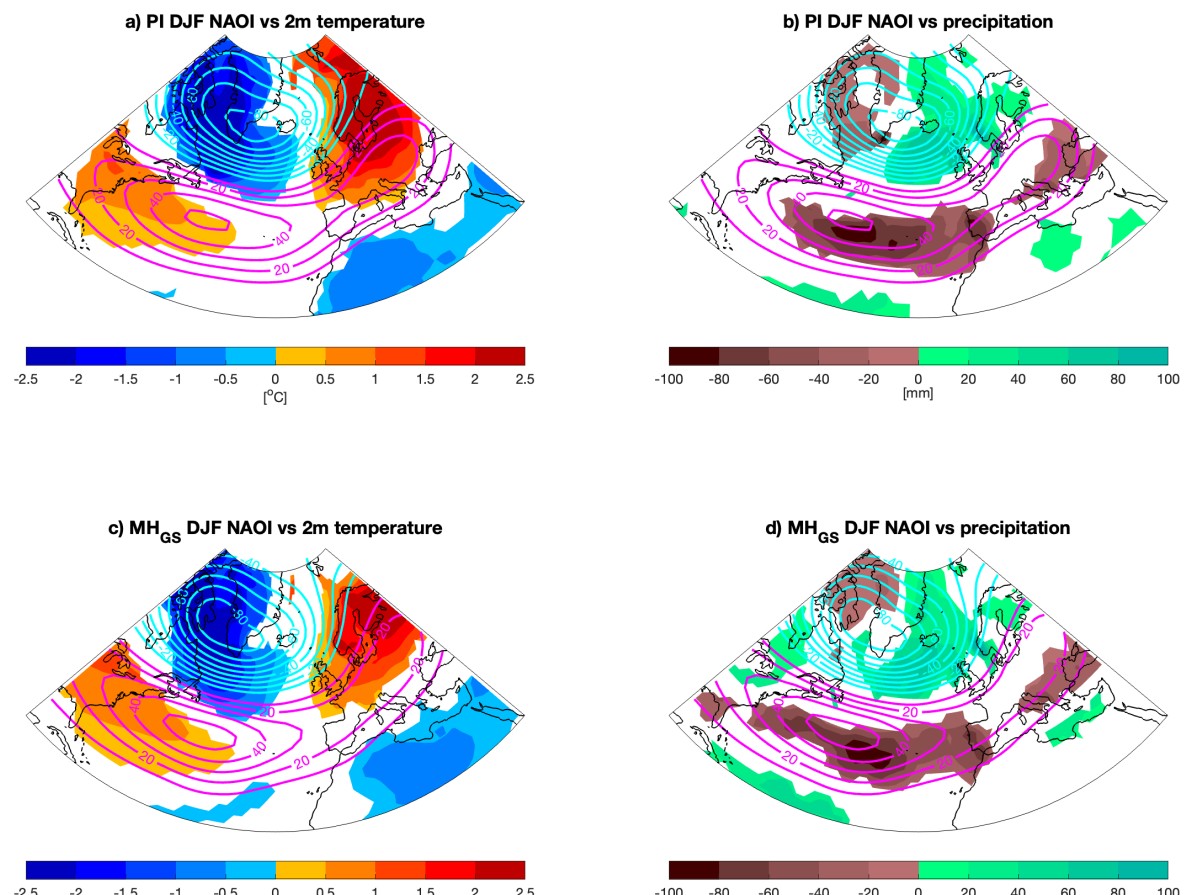

**Figure 7. Winter NAO patterns (contours) and associated thermal and rainfall anomalies (shadings), obtained by regressing, respectively, geopotential height at 500 hPa [m], (a, c) 2-m temperature and (b, d) precipitation onto the NAOI in the (a, b) PI and (c, d) MH$_{GS}$ simulations. Only significant anomalies in 2-m temperature and precipitation are shown, assessed by using a Student's *t* test at 95% level of confidence.**

In summer, the modelled NAO pattern reflects the canonical summer NAO pattern, characterised by weaker and less geographically extended anomalies than its winter counterpart, along with a northward shift of the meridional dipole (Bladé et al., 2012; Folland et al., 2009), explaining 23% of the total variability **(Fig. A3b)**. In the PI simulation, the NAOI is characterised by a prevailing positive phase **(Fig. 6b)**, associated with warm summers in western Europe and eastern North America and cold summers in the eastern Mediterranean and the North Atlantic, accompanied by dry conditions in northern Europe and wet conditions in the western Mediterranean **(Fig. 8a, b)**. The prevailing NAO phase turns negative when the



orbital parameters are changed **(Fig. 6b)**. The phase shift is statistically significant in both the MH$_{PMIP}$ and MH$_{GS}$ experiments with respect to the PI simulation, as verified by a Kolmogorov-Smirnov test (p<0.01). However, it is noteworthy that the Saharan greening effect does not introduce significant differences in this phase shift (p<0.49). A prevailing negative NAO phase then results in warmer summers in the eastern Mediterranean and northern Africa, and wetter summers in northern

Europe **(Fig. 8c, d)**.

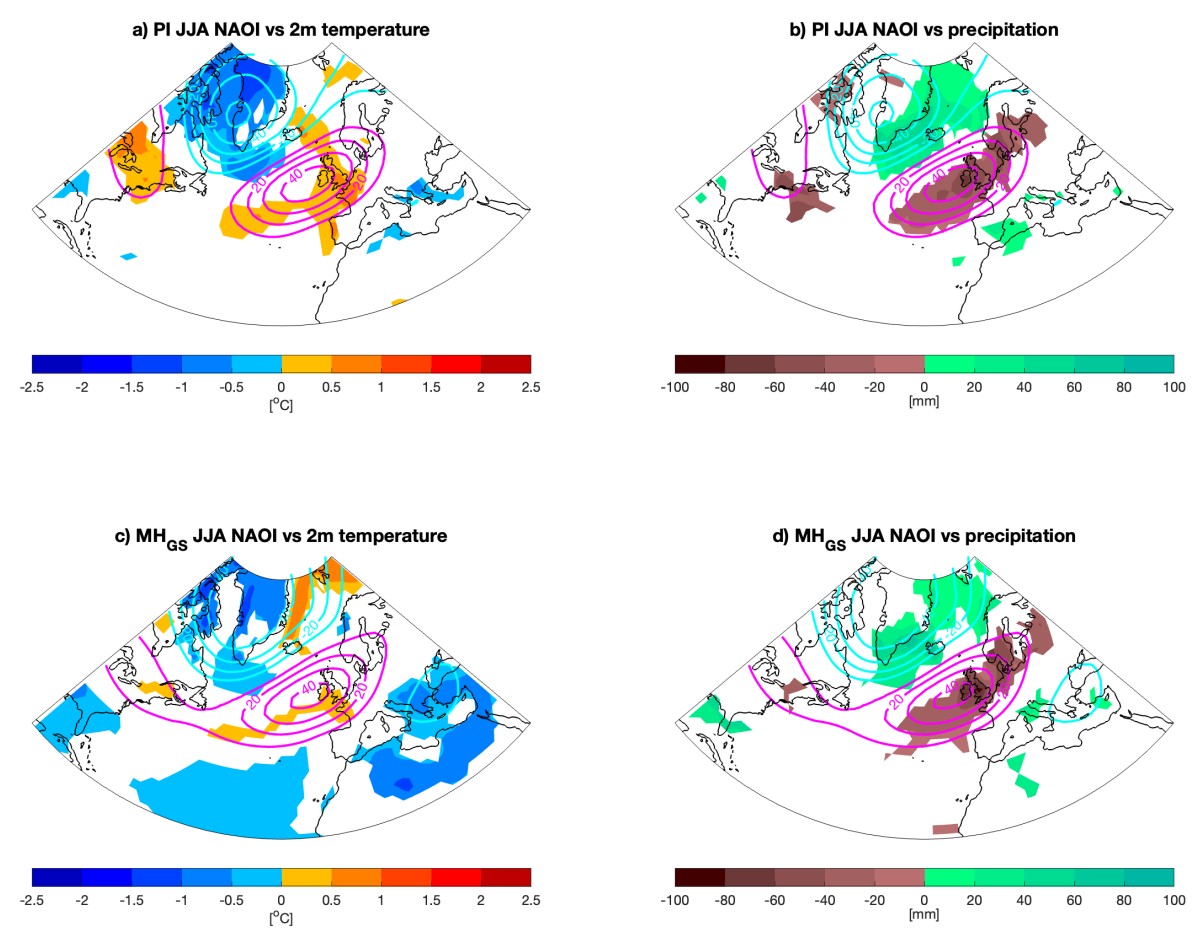

**Figure 8.** Summer NAO patterns (contours) and associated thermal and rainfall anomalies (shadings), obtained by regressing,

respectively, geopotential height at 500 hPa [m], (a, c) 2-m temperature and (b, d) precipitation onto the NAOI in the (a, b) PI and (c, d) MH$_{GS}$ simulations. Only significant anomalies in 2-m temperature and precipitation are shown, assessed by using a Student's *t* test at 95% confidence level.



### 3.3 North Atlantic Weather Regimes

The modifications in the atmospheric circulation variability over North America, the North Atlantic and Europe are explored at synoptic time scale through the analysis of the North Atlantic WRs (Michelangeli et al., 1995; Hochman et al., 2021) for the PI simulation and the MH experiments. Although year-round definitions of North Atlantic WRs exist (Hochman et al., 2021; Grams et al., 2017), they are often defined separately for the summer and winter seasons, and this is the approach adopted in this paper.

Model simulations show that the winter synoptic circulation is characterised by the four canonical WRs described in the literature (Cassou et al., 2004). The occurrences of these WRs are relatively uniform: two WRs are associated with the NAO's positive and negative phases (NAO+ and NAO-), accounting for approximately 50% of the analysed days. The remaining 50% are associated with a Scandinavian blocking (SB) or an Atlantic ridge (AR) pattern **(Table 2 and Fig. A4)**. It is highlighted that NAO+ and NAO-, as defined from the clustering of the daily variability, do not show the spatial symmetry typically

associated with the NAO's positive and negative phases defined as the first EOF of the interannual variability (see **Fig. A3a and A4a, b**). Consistent with the analysis of the monthly NAO and previous research (Nesje et al., 2001; Rimbu et al., 2003; Olsen et al., 2012), the modification of the orbital parameters leads to a reduction in the occurrence of NAO+, which becomes residual in the MH$_{GS}$ experiment with respect to the PI simulation, along with an increase in the frequency of NAO- from PI to MH$_{GS}$ **(Table 2)**. However, the increase in NAO- does not fully offset the decrease in NAO+, resulting in increased AR and

SB frequencies, with SB becoming the dominant WR (30.8%) in the MH$_{GS}$ experiment **(Table 2)**. The changes in the WR occurrence show a monotonic behaviour from the PI simulation to the MH experiments, with more pronounced changes observed in the MH$_{GS}$ experiment compared to MH$_{PMIP}$ **(Table 2)**, suggesting that the effect of the Saharan greening on the atmospheric circulation and the associated thermal and rainfall anomalies amplifies the changes driven solely by the orbital forcing. Therefore, the modifications in the WR dynamics are discussed in detail only for the MH$_{GS}$ experiment.

The NAO+ circulation pattern dominating the winter circulation in PI is associated with warm and wet anomalies in central and western Europe and southern Scandinavia, and cold and dry anomalies in North America and northern Scandinavia, along with dry anomalies in the eastern Mediterranean **(Fig. 9a and 10a)**. In the MH$_{GS}$ experiment, the large reduction in the occurrence of the NAO+ pattern is partially offset by the increased occurrence of NAO-, leading to cold anomalies in Scandinavia and eastern North America, paired with warm anomalies in polar North America, western Mediterranean and

North Africa. It also results in wet anomalies over polar North America and central and eastern Europe, while dry anomalies in the Mediterranean **(Fig. 9d and Fig. 10d)**. The increased occurrence of SB in the MH$_{GS}$ experiment indicates reinforced cold anomalies in polar North America, Europe and the Mediterranean, and warm anomalies in northern Scandinavia and eastern North America, accompanied by dry anomalies in western and central-northern Europe and polar North America, and wet anomalies in the Mediterranean **(Fig. 9f and Fig. 10f)**.






**Table 2. WR occurrence in the PI and MH simulations, in percentage. The regimes are: NAO+, positive phase of the North Atlantic Oscillation; NAO-, negative phase of the North Atlantic Oscillation; AR, Atlantic Ridge; SB, Scandinavian Blocking; IL, Icelandic Low.**

| | NAO+ | NAO- | AR | SB | IL |
|---|---|---|---|---|---|
| Winter | | | | | |
| Concatenated simulations | 26.8 | 23.0 | 24.8 | 25.4 | |
| PI | 40.2 | 17.6 | 23.1 | 19.1 | |
| MH$_{PMIP}$ | 26.7 | 23.1 | 25.7 | 24.4 | |
| MH$_{GS}$ | 13.4 | 28.3 | 27.5 | 30.8 | |
| Summer | | | | | |
| Concatenated simulations | 28.3 | 25.4 | 18.5 | | 27.8 |
| PI | 20.7 | 21.8 | 25.5 | | 32.1 |
| MH$_{PMIP}$ | 27.7 | 27.3 | 17.9 | | 27.1 |
| MH$_{GS}$ | 36.4 | 27.2 | 12.1 | | 24.3 |



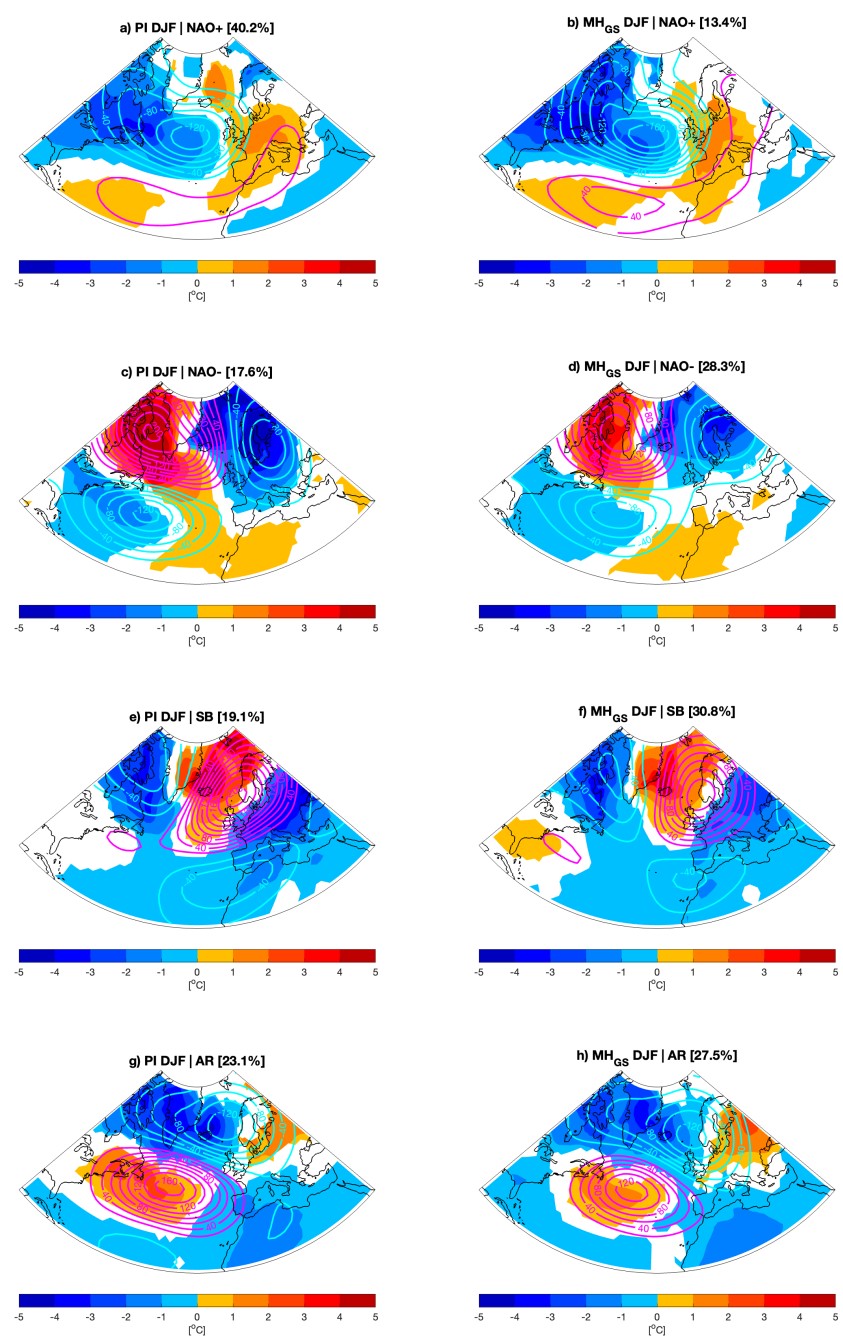


**Figure 9. Winter North Atlantic WRs and associated thermal anomalies, respectively defined as the anomalies of the geopotential height at 500 hPa [m] and the 2-m temperature with respect to the climatology, in the (left) PI and (right) MH$_{GS}$ simulations. Only significant anomalies in 2-m temperature are shown, assessed by using a Student's *t* test at 95% confidence level.**



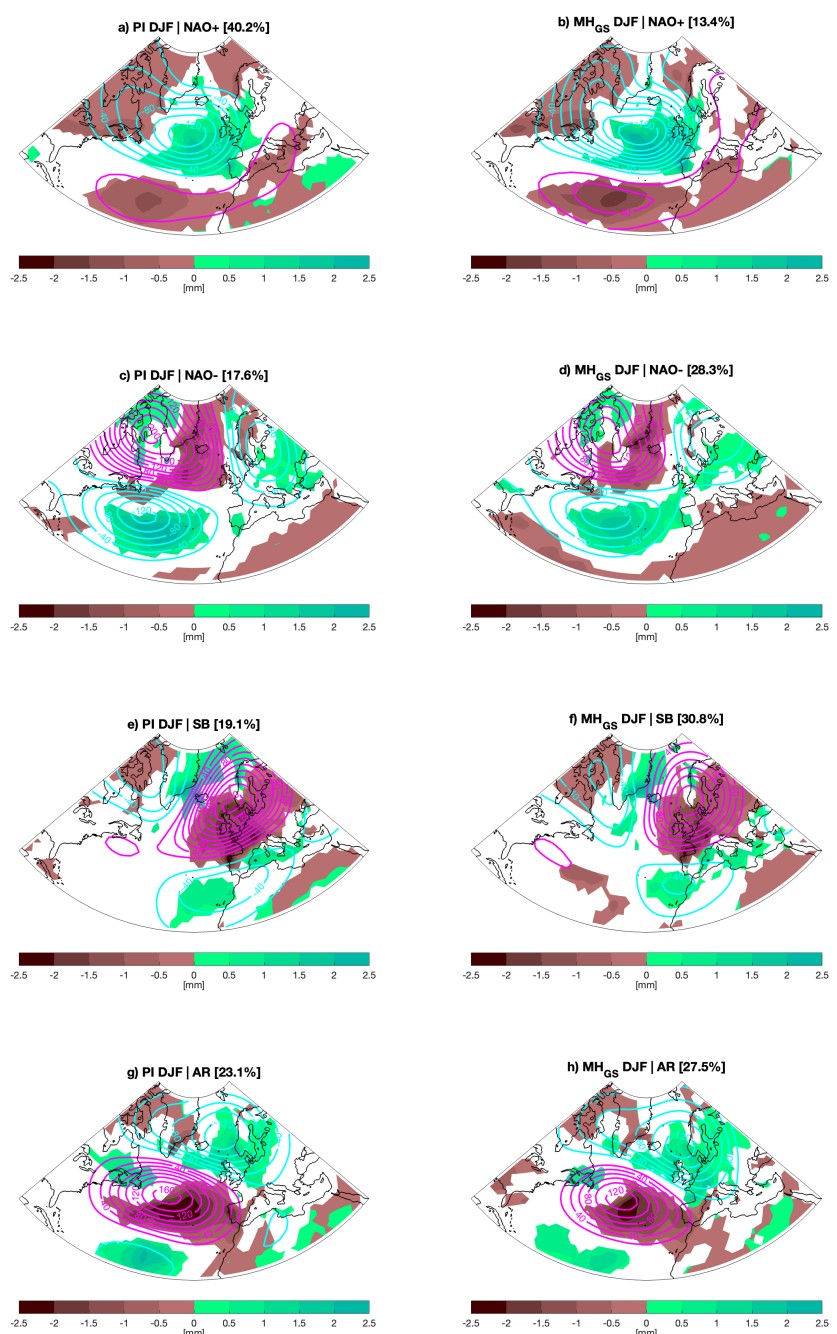

**Figure 10. Winter North Atlantic WRs and associated rainfall anomalies, respectively defined as the anomalies of the geopotential height at 500 hPa [m] and daily precipitation with respect to the climatology, in the (left) PI and (right) MH$_{GS}$ simulations. Only significant anomalies in precipitation are shown, assessed by using a Student's *t* test at 95% confidence level.**



Consistent with the literature, the summer synoptic circulation simulated by EC-Earth3.1 is characterised by NAO+, NAO-, AR and the Icelandic low (IL) (Cassou et al., 2005) **(Fig. A5)**. The NAO+, NAO- and IL show similar frequencies,

characterising the atmospheric circulation in more than 80% of the analysed daily fields, while the occurrence of AR is much lower **(Table 2)**. Notably, the summer NAO+ and NAO- do not display symmetrically opposite circulation, as seen in the NAO's positive and negative phases defined as the first EOF of the interannual variability (see **Fig. A3b and A5a, b**). The modification of the orbital parameters in the MH experiments leads to an increased occurrence of both NAO+ and NAO- with respect to PI, along with a decrease in the IL and AR frequencies, making the NAO the dominant pattern of the synoptic

variability in the $MH_{GS}$ (63.6% of the analysed daily fields are associated with NAO WRs, **Table 2**). The discrepancy between the increased occurrence of both NAO+ and NAO- WRs and the shift towards a negative phase in the monthly NAO can be explained by the differences in the spatial patterns discussed above. While NAO- well matches the negative phase of the monthly NAO (see **Fig. A3b and A5a**), NAO+ does not display a symmetric counterpart, with the high-pressure centre of action shifted above the Scandinavian peninsula (see **Fig. A5b**). This discrepancy is more a matter of terminology than a

physical inconsistency. The predominance of NAO+ and NAO- WRs results in warm anomalies affecting southern and northern Europe and eastern North America when the Saharan greening is prescribed **(Fig. 11b, d)**. Conversely, the precipitation anomalies associated to the WR shift towards a NAO-dominated circulation primarily affect Europe, with no significant impact on North America. Specifically, the NAO+ is associated with dry anomalies in Scandinavia and the eastern Mediterranean, while NAO- leads to dry anomalies in southern Europe and the Mediterranean, accompanied by wet anomalies

in western Europe and southern Scandinavia **(Fig. 12b, d).**





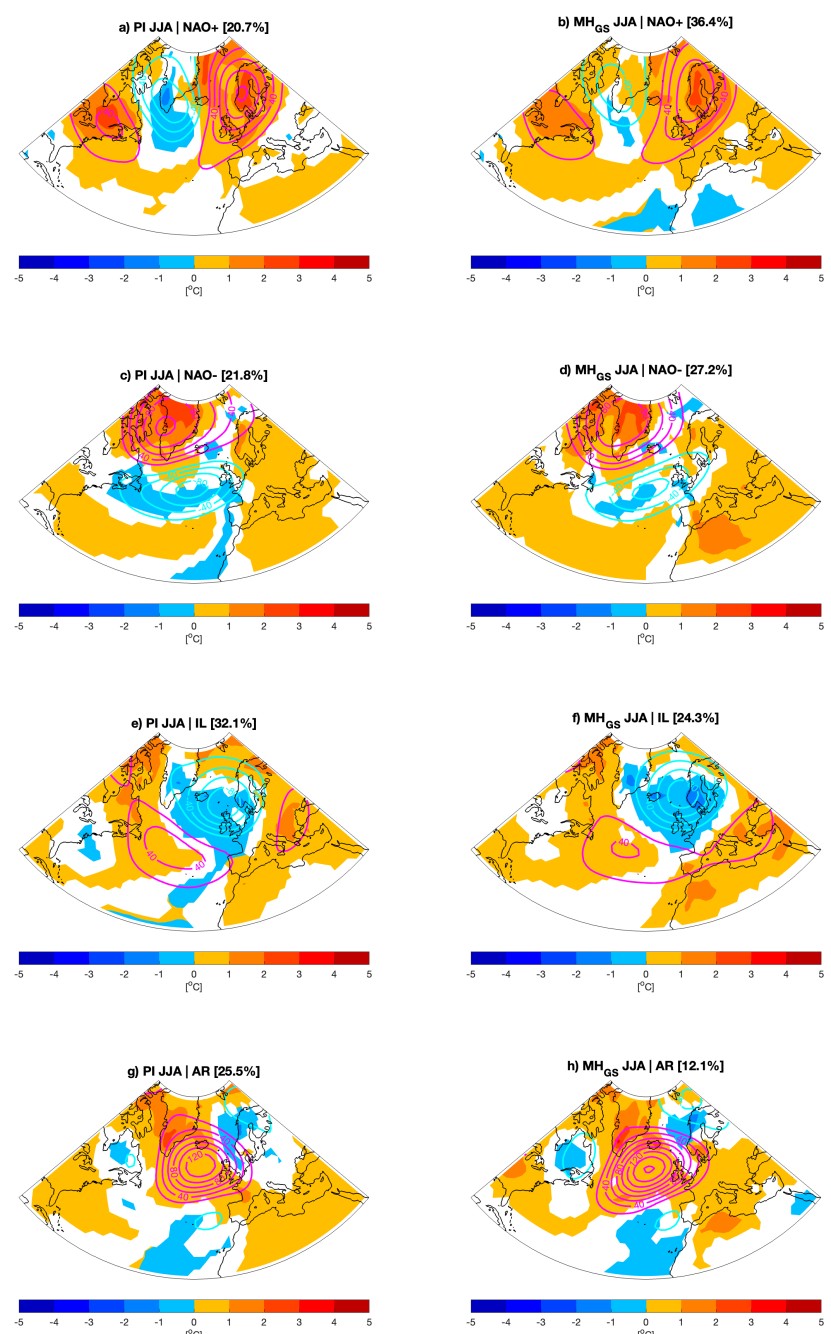

**Figure 11. Summer North Atlantic WRs and associated thermal anomalies, respectively defined as the anomalies of the geopotential height at 500 hPa [m] and the 2-m temperature with respect to the climatology, in the (left) PI and (right) MH$_{GS}$ simulations. Only significant anomalies in 2-m temperature are shown, assessed by using a Student's *t* test at 95% confidence level.**



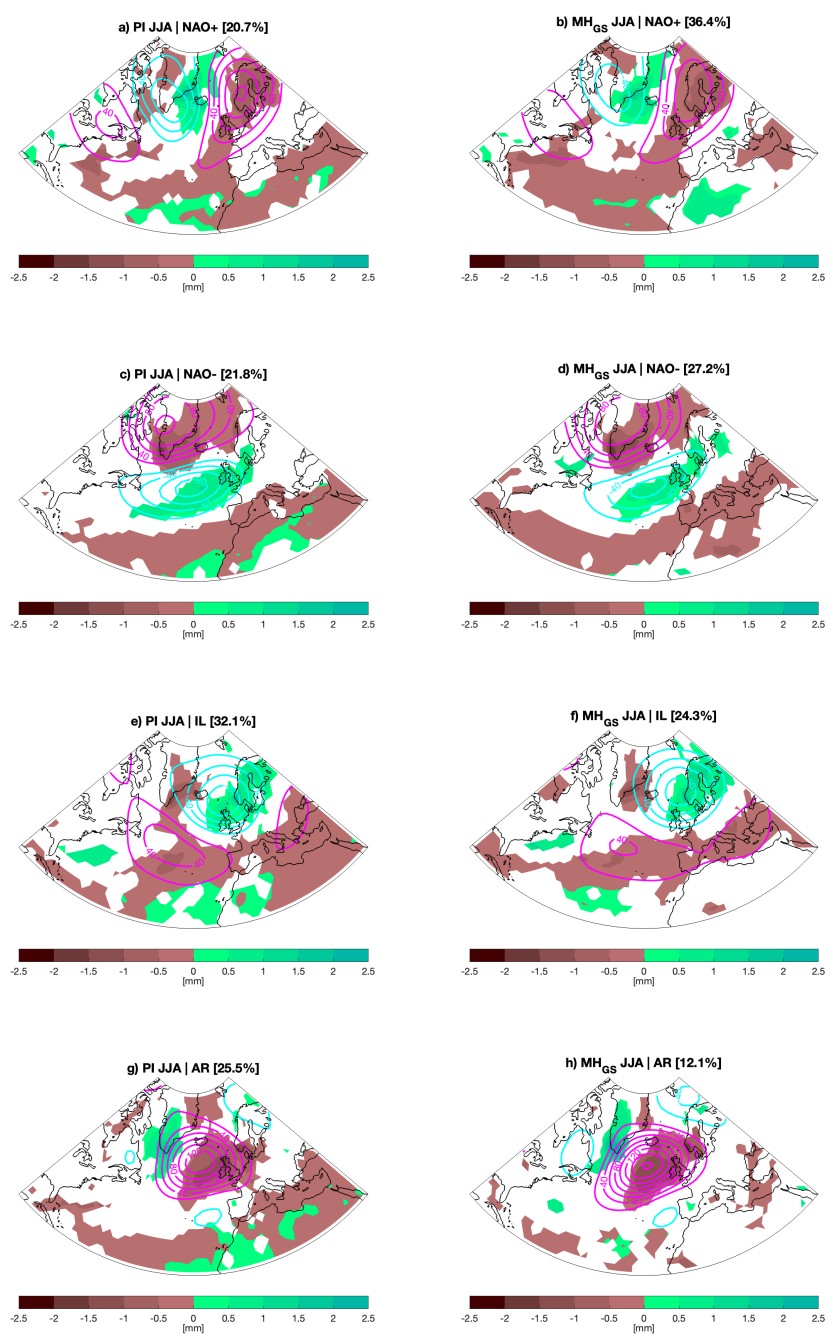

**Figure 12. Summer North Atlantic WRs and associated rainfall anomalies, respectively defined as the anomalies of the geopotential height at 500 hPa [m] and daily precipitation with respect to the climatology, in the (left) PI and (right) MH$_{GS}$ simulations. Only significant anomalies in precipitation are shown, assessed by using a Student's *t* test at 95% confidence level.**




## 4. Proxy-model comparison

The Cohen's Kappa index (κ) is used to quantify qualitative agreement between two datasets, with values ranging from -1 (complete disagreement) to 1 (perfect agreement). Here, the index indicates a generally low agreement between proxy reconstructions and model outputs over the mid-latitudes **(Table 3)**. Notably, the agreement for summertime temperatures is
particularly low, with κ values very close to zero over all regions. For wintertime temperatures, κ values range from 0 to 0.158, indicating very low agreement except for the $MH_{PMIP}$ simulation over Asia (κ = 0.158). Annual precipitation shows slightly higher κ values, peaking at 0.28 for the $MH_{GS}$ simulation over Asia. In all cases but one (annual precipitation over Asia), the $MH_{PMIP}$ simulation shows higher κ values compared to the $MH_{GS}$ simulation.

A closer inspection of the proxy-model comparison reveals several factors contributing to the low agreement in summer
temperatures. Firstly, the proxy reconstructions lack a spatial coherent large-scale pattern across the selected mid-latitude regions, with some coherence only found in some sub-regions, like Scandinavia. Secondly, both the $MH_{PMIP}$ and $MH_{GS}$ simulations indicate a strong mid-latitude warming signature, contrasting with the cooling shown in several reconstructions in North America and northern Asia. This suggests that the model may simulate an overly homogenous warming signal, while proxies indicate localised temperature increases. More importantly, the lack of coherent regional temperature signatures in the
proxy reconstructions, suggest that low proxy-model agreement may not be exclusively due to model deficiencies. For annual precipitation, consistent regional signatures emerge over North America and Asia. North American proxies suggest a drier MH in the west and east, but wetter conditions in the southwest. While both $MH_{PMIP}$ and $MH_{GS}$ simulations capture the slight wettening in the southwest, they fail to simulate the drying reconstructed elsewhere. In Asia, a minor drying from ~45° E to ~100° E in the $MH_{PMIP}$ simulation is replaced by more extensive wettening in the $MH_{GS}$ simulation, leading to an improvement
in the κ value for the region.

The overall low proxy-model agreement is further complicated by numerous inconclusive MH proxy records and the lack of consistent regional proxy signatures, except in specific sub-regions. This raises questions about the nature of MH climate anomalies: either the mid-latitudes lack coherent seasonal temperature patterns (unlike the tropics and high latitudes), suggesting limitations in the model's ability to capture regional climate nuances, or coherent climate signals do exist, and
improvement in proxy-model agreements depend more on resolving discrepancies between proxies rather than solely on model improvements. In summary, the inclusion of vegetated Sahara in the model leads to improved agreement over Asia for precipitation and realistic representation of the drying in North America, albeit some spatial inaccuracies. This suggests that the $MH_{GS}$ simulation more effectively captures precipitation patterns compared to seasonal temperature patterns across the mid-latitudes.




**Table 3.** Cohen's Kappa index for region-wise proxy-model comparison, following DiNezio and Tierney (2013).

|  | Temperature (DJF) |  | Temperature (JJA) |  | Annual precipitation |  |
|---|---|---|---|---|---|---|
| Region | $MH_{PMIP}$ | $MH_{GS}$ | $MH_{PMIP}$ | $MH_{GS}$ | $MH_{PMIP}$ | $MH_{GS}$ |
| North America: Pacific Coast | 0.007 | 0.0 | 0.0 | 0.0 | 0.161 | 0.072 |
| North America: Atlantic Coast | 0.03 | 0.0 | 0.0 | 0.0 | 0.138 | 0.025 |
| Western Europe and Mediterranean | 0.003 | 0.0 | 0.0 | 0.0 | 0.021 | 0.002 |
| Asia | 0.158 | 0.072 | 0.0 | 0.0 | 0.205 | 0.28 |

## 5. Discussion and Conclusions

In this study, a set of fully-coupled climate model simulations are analysed to explore the impact of Saharan greening on mid-
latitude atmospheric circulation and climate conditions in the Northern Hemisphere during the MH. Specifically, two MH
simulations are performed with and without prescribed vegetation cover and reduced dust emissions in the Sahara region, and
compared to a PI control experiment. The climatological response in the Northern Hemisphere mid-latitudes is analysed, along
with the modifications in the teleconnection patterns and the synoptic variability in the North Atlantic. To the authors'
knowledge, this is the first study attempting to assess the MH climate modifications, including the Saharan greening, at mid-
latitudes through the modelling of the atmospheric circulation variability at synoptic to interannual time scales.

The MH simulations show significant impacts on both surface temperature and precipitation in the mid-to-high latitudes. The
warming seen in the mid-to-high latitudes is attributed to increased summer insolation leading to decreased sea ice, with a
sustained effect into winter, known as "summer remnant effect of insolation" (Yin and Berger, 2012). The increase in
precipitation aligns with a significant reshaping of the large-scale circulation in the upper and middle troposphere, including a
westward shift of the global Walker Circulation and modifications in the westerly flow. Notably, these shifts result in altered
surface climate in North America and Eurasia. The responses in temperature, precipitation and atmospheric dynamics are more
pronounced in the $MH_{GS}$ simulation, indicating the significant influence of the Saharan greening on climate in the Northern
Hemisphere. There is a broad literature on tropical-extratropical interactions triggered by tropical forcings such as the Indian
monsoon and the El Niño/Southern Oscillation [e.g. Hoskins and Ambrizzi (1993); Rodwell and Hoskins (1996)]. More
recently, the African monsoon has been indicated as a possible source of tropical-extratropical teleconnections (Gaetani et al.,
2011; Nakanishi et al., 2021), reinforcing the hypothesis that the strengthening of deep convection in northern Africa associated
with the Saharan greening could lead to climate impacts in the extratropics. Moreover, the fact that the warm anomalies are
larger in $MH_{GS}$ than in $MH_{PMIP}$ points to other drivers, beside the insolation, that could have amplified the warming, such as





modifications in the ocean circulation. Studies investigating potential ocean circulation changes associated to the Saharan greening are hence important to better understand the widespread warming seen at mid-to-high latitudes.

The analysis of the interannual variability of the simulated mid-tropospheric circulation in the North Atlantic shows a significant shift from a prevailing positive NAO phase in the PI experiment to a prevailing neutral-to-negative phase in the MH experiments in both winter and summer. The impact is stronger when Saharan vegetation is prescribed, particularly in winter. The simulated changes in the NAO are in agreement with the existing literature (see e.g., Nesje et al., 2001; Rimbu et

al., 2003; Olsen et al., 2012). However, although the simulated positive-to-negative shift of the monthly NAO is consistent with the reconstructions of overall colder and dryer conditions in North America, it does not provide an explanation of the reconstructed thermal anomalies in Europe, especially regarding the warmer conditions in Scandinavia.

In this respect, the analysis of the North Atlantic WR dynamics helps reconcile this discrepancy. The spatial patterns of the NAO WRs at the synoptic time scale, particularly the summer NAO+ and winter NAO-, display centres of action located over

Scandinavia, differently from the interannual NAO patterns. In addition, it is shown that the simulated Saharan greening drives modifications in the occurrence of other modes of the synoptic variability. The increased frequency of SB in winter and NAO+ in summer, both characterised by warm anomalies over Scandinavia, aligns with the warmer conditions found in the proxy records in the region and suggests a link to Saharan greening.

The proxy-model comparison, while revealing limited agreement due to regional inconsistencies in proxy records, suggests

that, where coherent climate reconstructions exist, the changes driven by the Saharan greening in large-scale circulation indicate plausible explanations for the proxy evidence. This points to new opportunities for understanding the MH climate in the mid-latitudes. Furthermore, this modelling exercise also highlights the need for more refined MH climate modelling, such as prescribing realistic vegetation across latitudes and considering the seasonal vegetation cycle, to account for large- and local-scale climate feedbacks.






**Appendix**

**Table A1. List of the locations, with coordinates, and references of the proxy records used for the quantification of the proxy-model agreement.**

| Site name | Latitude | Longitude | Original source |
|---|---|---|---|
| **Annual precipitation** | | | |
| Oro Lake, Canada | 49.78 | -105.33 | Michels et al. (2007) |
| Ammersee | 48 | 11.12 | Czymzik et al. (2013) |
| Path Lake | 43.87 | -64.93 | Neil et al. (2014) |
| Neor, Iran | 37.96 | 48.56 | Sharifi et al. (2015) |
| Lake Van | 38.4 | 43.2 | Chen et al. (2008) |
| Aral Sea | 45 | 60 | Chen et al. (2008) |
| Issyk-Kul | 42.5 | 77.1 | Chen et al. (2008) |
| Wulun Lake | 47.2 | 87.29 | Chen et al. (2008) |
| Bosten Lake | 42 | 87.02 | Chen et al. (2008) |
| Bayan Nuur | 50 | 94.02 | Chen et al. (2008) |
| Hovsgol Nuur | 51 | 101.2 | Chen et al. (2008) |
| Juyan Lake | 41.8 | 101.8 | Chen et al. (2008) |
| Gun Nuur | 50.25 | 106.6 | Chen et al. (2008) |
| Hulun Nuur | 48.92 | 117.38 | Chen et al. (2008) |
| Achit Nuur, Mongolia | 49.42 | 90.52 | Sun et al. (2013) |
| Oro Lake, Canada | 49.78 | -105.33 | Michels et al. (2007) |
| Cleland Lake, British Columbia | 50.83 | -116.39 | Steinman et al. (2016) |
| Paradise Lake, British Columbia | 54.685 | -122.617 | Steinman et al. (2016) |
| Lime Lake, Washington | 48.874 | -117.338 | Steinman et al. (2016) |



| Summer temperature | | | |
|---|---|---|---|
| Boothia Peninsula, Nunavut | 69.9 | -95.07 | Zabenski and Gajewski (2017) |
| North Lake | 69.24 | -50.03 | Axford et al. (2013) |
| Toskaljavri | 69.2 | 21.47 | Seppa et al. (2009) |
| KP2 | 68.8 | 35.32 | Seppa et al. (2009) |
| Myrvatn | 68.65 | 16.38 | Seppa et al. (2009) |
| Austerkjosen | 68.53 | 17.27 | Seppa et al. (2009) |
| Yarnyshnoe | 69.07 | 36.07 | Seppa et al. (2008) |
| Lapland | 69 | 25 | Helama et al. (2012) |
| 2005-804-004 | 68.99 | -106.57 | Ledu et al. (2010) |
| Liltlvatn | 68.52 | 14.87 | Seppa et al. (2008) |
| Gammelheimvatnet | 68.47 | 17.75 | Seppa et al. (2008) |
| Tsuolbmajavri | 68.41 | 22.05 | Seppa et al. (2008) |
| Lyadhej-To | 68.25 | 65.79 | Andreev et al. (2005) |
| Chuna | 67.95 | 32.48 | Solovieva et al. (2005) |
| Sjuuodjijaure | 67.37 | 18.07 | Rosen et al. (2001) |
| Kharinei | 67.36 | 62.75 | Jones et al. (2011) |
| MD95-2011 | 67 | 7.6 | Calvo et al. (2002) |
| MD99-2269 | 66.85 | -20.85 | Justwan et al. (2008) |
| B997-321 | 66.53 | -21.5 | Smith et al. (2005) |
| Berkut | 66.35 | 36.67 | Ilyashuk et al. (2005) |
| Iglutalik | 66.14 | -66.08 | Kerwin et al. (2004) |
| Screaming Lynx | 66.07 | -145.4 | Clegg et al. (2011) |
| Honeymoon Pond | 64.63 | -138.4 | Cwynar et al. (1991) |



| Svartvatnet | 63.35 | 9.55 | Seppa et al. (2009) |
|---|---|---|---|
| Tiavatnet | 63.05 | 9.42 | Seppa et al. (2009) |
| Kinnshaugen | 62.02 | 10.37 | Seppa et al. (2009) |
| Ratasjoen | 62.27 | 9.83 | Velle et al. (2005) |
| Hudson | 61.9 | -145.67 | Clegg et al. (2011) |
| Haugtjern | 60.83 | 10.88 | Seppa et al. (2009) |
| Holebudalen | 59.83 | 6.98 | Seppa et al. (2009) |
| Brurskardstjorni | 61.42 | 8.67 | Velle et al. (2005) |
| Moose Lake | 61.37 | -143.6 | Clegg et al. (2010) |
| Upper Fly Lake | 61.07 | -138.09 | Bunbury and Gajewski (2009) |
| Trettetjorn | 60.72 | 7 | Bjune et al. (2005) |
| Rainbow | 60.72 | -150.8 | Clegg et al. (2011) |
| s53s52 | 59.89 | -104.21 | Tillman et al. (2010) |
| Isbenttjonn | 59.77 | 7.43 | Seppa et al. (2009) |
| Flotatjonn | 59.67 | 7.55 | Seppa et al. (2009) |
| Grostjorn | 58.53 | 7.73 | Seppa et al. (2009) |
| Oykjamyrtjorn | 59.82 | 6 | Bjune et al. (2005) |
| LO09 | 58.94 | -30.41 | Berner et al. (2008) |
| K2 | 58.73 | -65.93 | Fallu et al. (2005) |
| Reiarsdalsvatnet | 58.32 | 7.78 | Seppa et al. (2009) |
| Dalene | 58.25 | 8 | Seppa et al. (2009) |
| Rice | 48.01 | -101.53 | Shuman and Marsicek (2016) |
| Steel | 46.97 | -94.68 | Shuman and Marsicek (2016) |
| Moon Lake | 46.86 | -98.16 | Shuman and Marsicek (2016) |



| | | | |
|---|---|---|---|
| Pickerel | 45.51 | -97.27 | Shuman and Marsicek (2016) |
| Nutt Lake | 45.21 | -79.45 | Shuman and Marsicek (2016) |
| Graham Lake | 45.19 | -77.36 | Shuman and Marsicek (2016) |
| Mansell | 45.04 | -68.73 | Shuman and Marsicek (2016) |
| Sharkey | 44.59 | -93.41 | Shuman and Marsicek (2016) |
| High Lake | 44.52 | -76.6 | Shuman and Marsicek (2016) |
| Devils Lake | 43.42 | -89.73 | Shuman and Marsicek (2016) |
| Okoboji Lake | 43.37 | -95.15 | Shuman and Marsicek (2016) |
| Hams | 43.24 | -80.41 | Shuman and Marsicek (2016) |
| Sutherland | 41.39 | -74.2 | Shuman and Marsicek (2016) |
| Spruce Pond | 41.24 | -74.18 | Shuman and Marsicek (2016) |
| Chatsworth | 40.68 | -88.34 | Shuman and Marsicek (2016) |
| Hinterburgsee | 46.72 | 8.07 | Heiri et al. (2003) |
| Gemini Inferiore | 44.39 | 10.05 | Samartin et al. (2017) |
| Lago Verdarolo | 44.36 | 10.12 | Samartin et al. (2017) |
| Lago dell'Accesa | 42.99 | 10.9 | Finsinger et al. (2010) |
| Tagus Mud Patch | 38.6 | -9.5 | Rodriguez et al. (2009) |
| **Winter temperature** | | | |
| Dalmutladdo | 69.17 | 20.72 | Bjune et al. (2004) |
| Candelabra | 61.68 | -130.65 | Cwynar and Spear (1995) |
| Hail | 60.03 | -129.02 | Cwynar and Spear (1995) |
| IOW225517 | 57.7 | 7.1 | Emeis et al. (2003) |
| Lago dell'Accesa | 42.99 | 10.9 | Finsinger et al. (2010) |
| M25/4-KL11 | 36.7 | 17.7 | Emeis et al. (2003) |




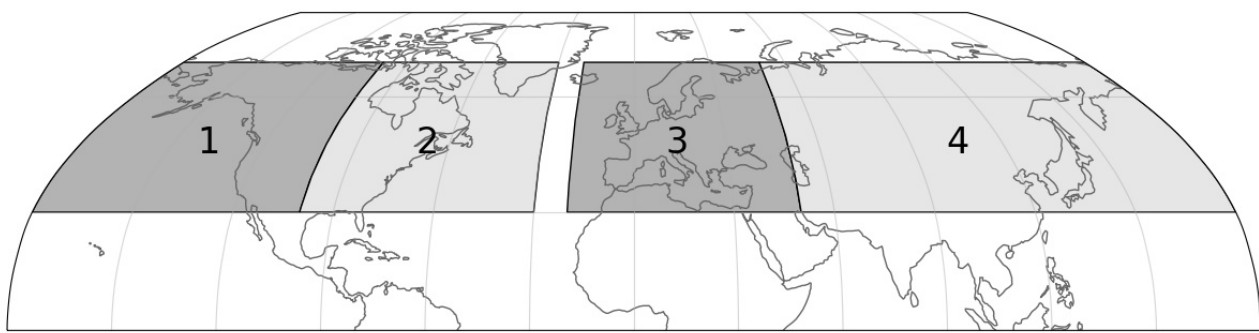

**Figure A1. Regions used for Cohen's Kappa index calculations. From left to right: (1) North America, Pacific Coast (180–100°W, 30–70°N), (2) North America, Atlantic Coast (100–30°W; 30–70°N), (3) Europe and Mediterranean (20°W–50°E, 30–70°N), and (4) Asia (50–180°E, 30–70°N).**


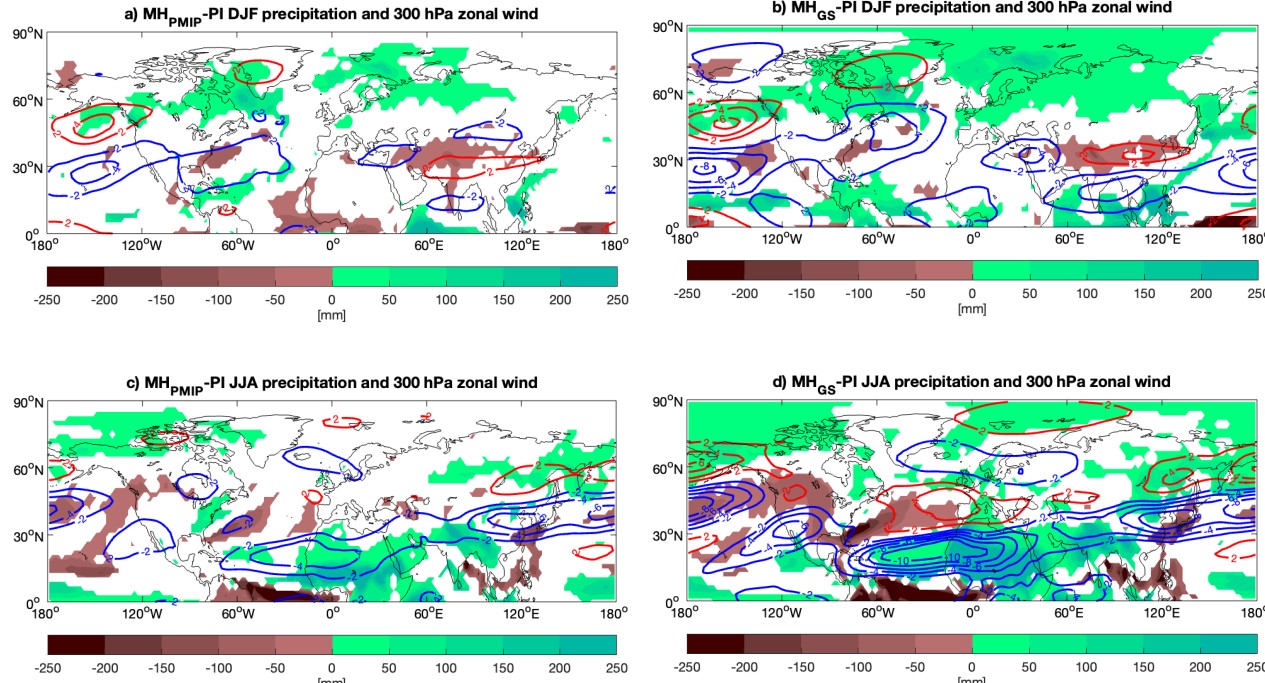

**Figure A2. Changes in precipitation [mm day$^{-1}$] in the (a, c) MH$_{PMIP}$ and (b, d) MH$_{GS}$ simulations with respect to the PI simulation, in (a, b) winter and (c, d) summer. Only areas showing statistically significant precipitation anomalies, estimated by a Student's t-test at 95% confidence level, are shown. Red and blue contours show, respectively, positive and negative change in the zonal wind velocity at 300 hPa [m s$^{-1}$].**




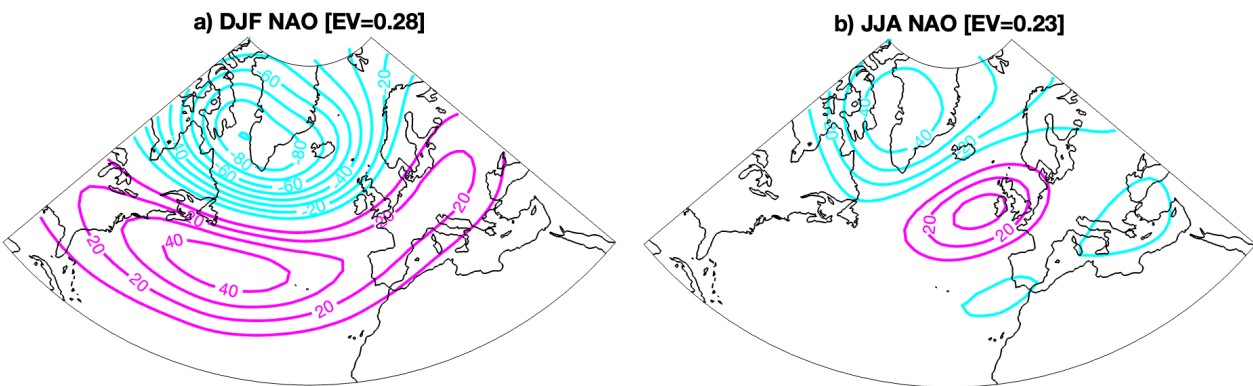

**Figure A3. NAO anomaly patterns in (a) winter and (c) summer, obtained by regressing geopotential height at 500 hPa [m] onto the**
**NAOI.**

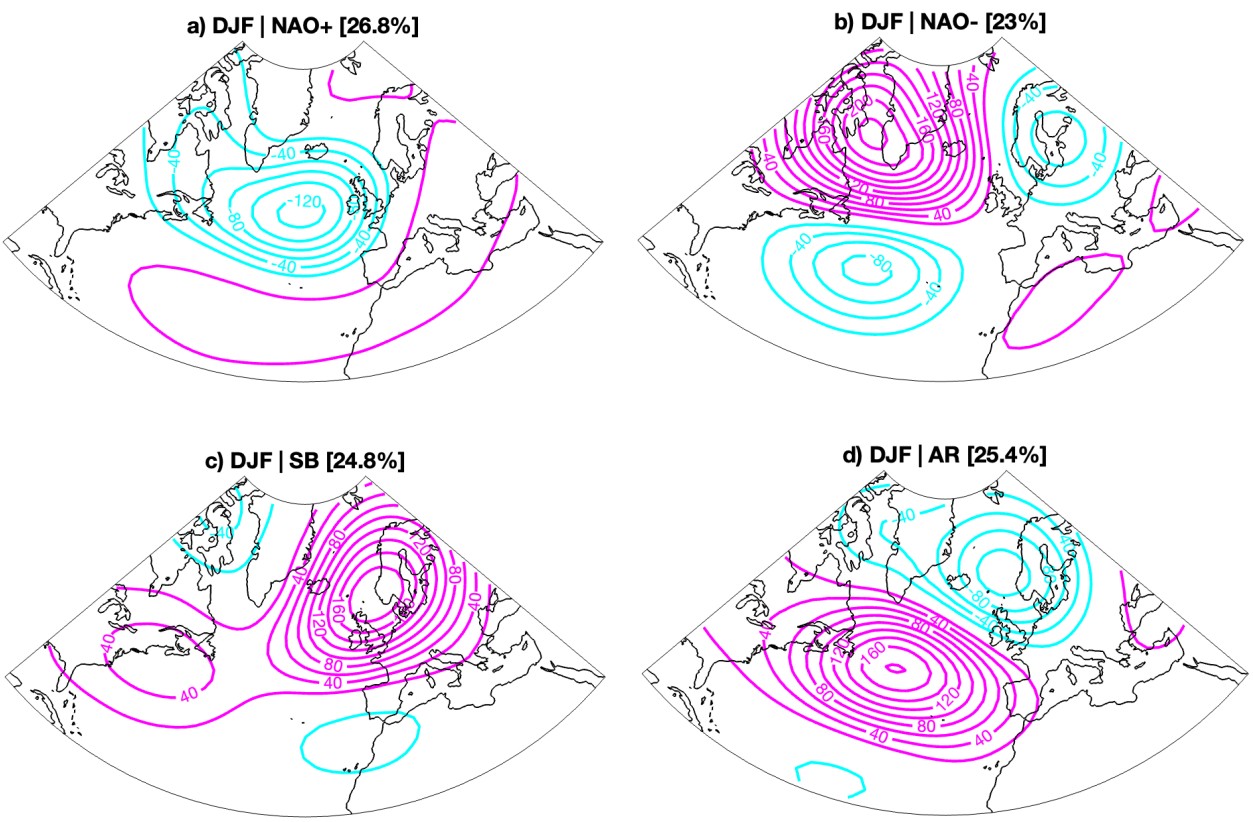

**Figure A4. Winter North Atlantic WRs, defined as the anomalies of the geopotential height at 500 hPa [m] with respect to the**
**climatology.**



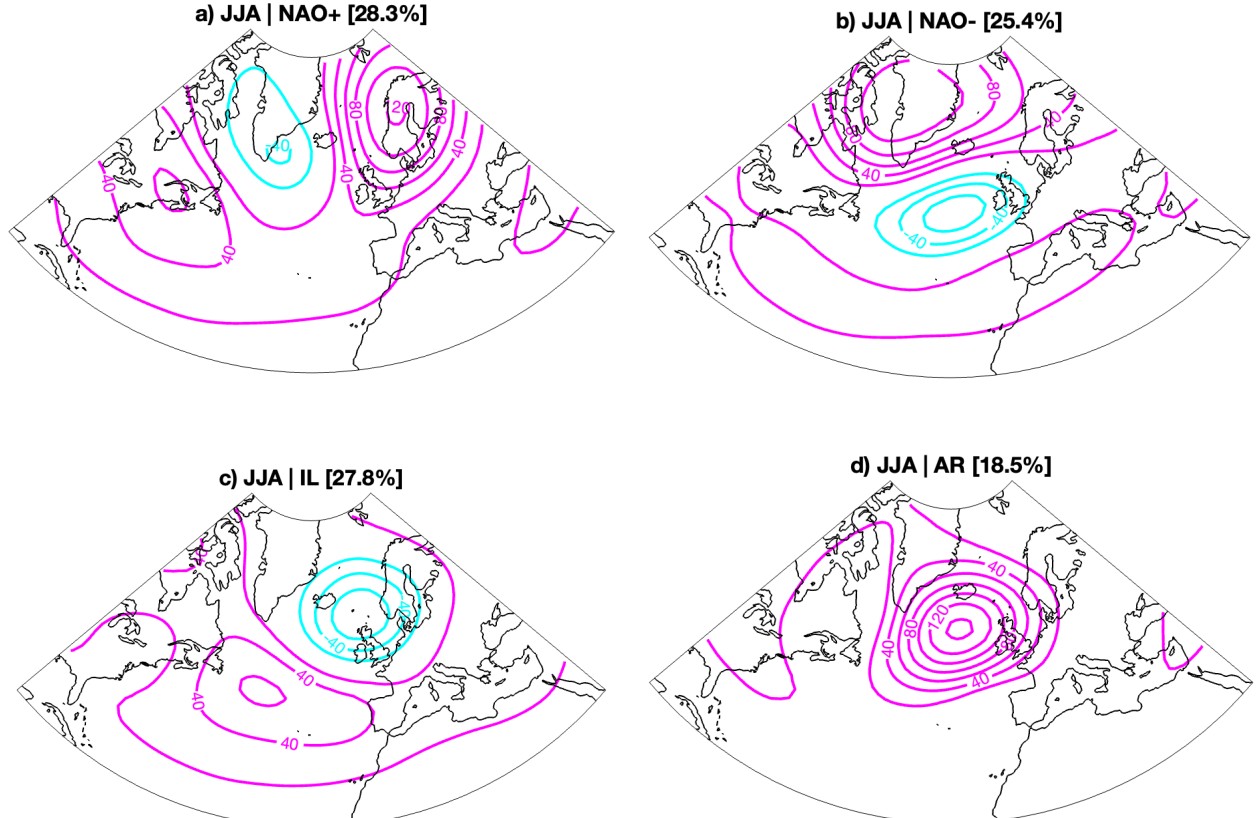

**Figure A5. Summer North Atlantic WRs, defined as the anomalies of the geopotential height at 500 hPa [m] with respect to the climatology.**

**Code and data availability**

All data and code are available upon request.

**Author contribution**

MG, GM and FSRP conceived the study. MG analysed the model simulations and wrote the paper. ST performed the proxy-model comparison and wrote the related section. MCAC and MG performed the WR classification. QZ run the simulations. All the authors commented on the manuscript.

**Competing interests**

The authors declare that they have no conflict of interest



**Acknowledgements**

The authors thank S. Harrison for useful discussion. MG acknowledge the support of the project "Dipartimenti di Eccellenza
2023-2027", funded by the Italian Ministry of Education, University and Research at IUSS Pavia; and of the International
Meteorological Institute based at the Department of Meteorology of Stockholm University. GM acknowledges the support of
the Department of Earth Sciences of Uppsala University. F.S.R.P. and S.T. acknowledge the financial support from the Natural
Sciences and Engineering Research Council of Canada (grant RGPIN-2018-04,981) and NSERC-FRQNT NOVA (grants
ALLRP 577112-22 and 2023-NC-324826).

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
