# Peer review of "Mid-Holocene climate at mid-latitudes: assessing the impact of the Saharan greening"

_EGUsphere, 2024_

## Author Comment (AC1)

**Response to the Editor's comments**

Firstly, we thank Prof. Claussen for considering our manuscript fitting the scope of the journal and advancing it for peer-review stage. We take here the opportunity to address his insightful comments as follows:

The term "fully coupled model" may indeed give the impression of a more extensive coupling than is present in our simulations. The model we used is an atmosphere-ocean coupled model without dynamic interaction with vegetation, lakes, soil and carbon cycle. We have revised the terminology in the manuscript to reflect this more accurately, now referring to it as an "atmosphere-ocean coupled climate model".

With respect to the sensitivity of the EC-Earth model to dust effect, we first highlight that, although including dust reduction does increase simulated rainfall by about 30%, most of the monsoon strengthening observed in our simulations is attributed to the effects of Sahara greening (see Pausata et al. 2016). Furthermore, a strong impact from dust reduction in the strengthening of the African monsoon during the MH has been reported by Thompson et al. (2019), showing a contribution of about 15%–20% to the total increase in rainfall. Hopcroft and Valdes (2019) discussed the variability in dust-precipitation feedbacks dependent on dust optical properties and particle size with precipitation changes ranging from -20% to +50%. Similarly, Sagoo and Storelvmo (2017) have shown significant rainfall changes simulated for idealized low and high dust concentrations. These studies support the robustness of our results against the model sensitivity to dust effect. We add a sentence in Section 2 to address this aspect:

"The inclusion of dust reduction in the experimental setup significantly influences the simulation of the monsoonal dynamics, leading to an increase of around 30% of cumulated precipitation compared to simulations with prescribing vegetation only (see Pausata et al., 2016). Similar impact has been observed in other modelling studies by Thompson et al. (2019), Hopcroft and Valdes (2019) and Sagoo and Storelvmo (2017), confirming the relevance and comparability of the model's response to dust effects".

Pausata, F. S. R., Messori, G., and Zhang, Q.: Impacts of dust reduction on the northward expansion of the African monsoon during the Green Sahara period, Earth Planet Sci Lett, 434, 298–307, https://doi.org/10.1016/j.epsl.2015.11.049, 2016.

Thompson, A. J., Skinner, C. B., Poulsen, C. J., & Zhu, J. (2019). Modulation of mid-Holocene African rainfall by dust aerosol direct and indirect effects. Geophysical Research Letters, 46, 3917–3926. https://doi.org/10.1029/2018GL081225

Hopcroft, P. O., & Valdes, P. J. (2019). On the role of dust-climate feedbacks during the mid-Holocene. Geophysical Research Letters, 46, 1612–1621. https://doi.org/10.1029/2018GL080483

Sagoo, N., and T. Storelvmo (2017), Testing the sensitivity of past climates to the indirect effects of dust, Geophys. Res. Lett., 44, 5807–5817, doi:10.1002/2017GL072584.

**Response to R1**

The authors analyse mid Holocene climate model simulations with and without taking into account the effects of prescribed Saharan greening (SG). The model experiments using EC-Earth have been described elsewhere, and the new aspect of this paper is the focus on mid-latitude atmospheric circulation patterns and changes in weather regimes (WR). The authors discriminate between changes induced by the orbital and GHG forcing as in the standard PMIP3 simulation, and additional modifications of climate features by including prescribed SG.

The topic of the manuscript addresses a relevant scientific question in the discussion of causes for past climate changes on the regional and continental level, where a lot of discrepancy remains between proxy reconstructions and model simulations.

Extending the analyses to the mid-latitudes is therefore timely and useful and provides important insights.

The study underlines the importance of including changes in vegetation and the related dust fields not only for the monsoon systems, but also for mid-latitude circulation and circulation patterns. The authors find significant effects coming from SG for temperature and precipitation fields. Regarding variability patterns, the effect of SG is merely an enhancement of effects that are already seen in the standard PMIP experiments, although they are not making an attempt to explain why the SG effects would do that.

Another important finding is that one has to include variability on the daily timescale to refine the regional effects by means of WRs, which sometimes give different patterns than circulation patterns based on monthly data. The authors also discuss if and how much the proxy-data gap can be narrowed down in the SG simulation, but with rather limited success.

Overall, the manuscript is written in a clear and concise way and the findings are well supported by the prescribed analyses and figures.

Save for a few inconsistencies, I find the manuscript quite mature and I would recommend that the paper can be published after (very) minor revisions.

We thank the reviewer for the time s/he spent in revising our work and her/his positive reception, as well as for the comments which will be useful to improve the manuscript.

**Minor issues:**

Abstract, ln 21ff: Don't you say in the paper that the NAO+/- shift comes about in the run without SG and that SG just enhances this effect (c.f., Fig. 6)?

Thanks for highlighting this oversimplifying phrasing. From figure 6 we see that monthly NAO shits from prevailing positive in PI to prevailing negative in both the sensitivity experiments in summer, while it shifts to negative in the SG experiment and neutral in the no-SG experiment. We rephrase to clarify:

"Furthermore, the Saharan greening modifies the atmospheric synoptic circulation over the North Atlantic, enhancing the effect of the orbital forcing on the transition of the North Atlantic Oscillation phase from prevailingly positive to negative in winter and summer".

Figure 1: blue dots, in particular over southern Europe are hard to distinguish from the gray ones, maybe make them slightly bigger?

The figure is now improved, by making the grey dots lighter, to make blue and red dots more visible.

Ln 99: how uncertain is this estimate?

"This vegetation cover leads to an average decrease in surface albedo from 0.30 during the PI period to 0.15".

There's no uncertainty in the surface albedo value, because this is part of the vegetation prescription in the simulation, that is changing the desert albedo (0.30) to shrub albedo (0.15). We modify the sentence for clarification:

"This modification in land surface type leads to a change of surface albedo from 0.30 (desert) to 0.15 (shrub)".

Ln 115: a few more words about the Cohen and how it is applied over the selected regions should be included so readers don't have to look it up.

Thanks for this comment. A short description of the computation of the Cohen index is now added:

"The Cohen's Kappa index quantifies agreement between climate variables form climate simulations and proxy reconstructions through the probability that, at the proxy sites, the two data sets agree on the category of anomaly (e.g., positive/negative/no change in both simulation and proxy), but not by chance alone. The index is calculated by constructing a data matrix with the number of sites where the two data

sets agree, partially disagree (one indicates a positive or a negative anomaly, while the other indicates no change) and completely disagree (one indicates a positive anomaly while the other indicates a negative anomaly, and vice versa). This data matrix is then multiplied by a weight matrix, to penalise complete disagreement more than partial disagreement. The resultant values for the index range from 0 to 1, where 0 indicates no agreement, 0.5 indicates partial agreement, and 1 indicates perfect agreement".

Ln 295: maybe include also a discussion how that relates to the suggestions by Mauri et al. (Clim Past, 2014)

Many thanks for suggesting this interesting reference, that we missed. The comparison with Mauri et al. 2014 is now discussed in Section 5:

"The circulation anomalies associated with SB in winter and NAO+ in summer are consistent with those suggested by Mauri et al. (2014) to explain the MH thermal and precipitation anomalies in Northern Europe, namely stronger westerly and southerly flows towards Scandinavia in winter and summer".

Ln 381: "impacts" of what?

The impact of simulating with and without SG. Now rephrased:

"The MH experiments show significant changes in both surface temperature and precipitation in the mid-to-high latitudes with respect to the PI control simulation".

Ln 394: what gives you the idea of looking into changing ocean circulation, any references for that?

We did not look into the ocean circulation indeed; this was beyond the intended scope of this study. Our considerations come from the distinct temperature response noted in our simulations, specifically the more pronounced warming in the MHGS simulation compared to MHPMIP (see Fig. 2), particularly in the North Atlantic (see Fig. 1), which suggests that the AMOC might respond to changes induced by the Sahara greening, potentially influencing the atmospheric circulation and temperature. We rephrase to clarify:

"Moreover, the fact that the warm anomalies are more pronounced in MHGS than in MHPMIP simulation points to other drivers, beside the insolation, that could have amplified the warming. In particular, the prominent warming in the North Atlantic simulated when vegetation is prescribed in the Sahara raises the possibility of modifications in the ocean circulation (see e.g. Zhang et al. 2021, who found a strengthening of the Atlantic Meridional Overturning Circulation in response to the simulation of a green Sahara). Such changes could feasibly feedback on the atmosphere. Consequently, further studies focused on investigating potential changes in ocean circulation associated with the Saharan greening would be valuable for a better understanding of the widespread warming seen at mid-to-high latitudes".

---

## Author Comment (AC2)

**Summary**

The author analyse two mid-Holocene climate model simulations using EC-Earth. The first of these is a standard PMIP type setup and the second includes a reduction in Saharan dust along with a prescribed greening of the Sahara. The authors focus on mid-latitude climatic impacts and in particular the changes in weather regimes and the NAO. The authors also evaluate the simulations with proxy records for temperature and precipitation. They find significant but similar impacts on the NAO and WR independent of the greening, but the agreement with proxy records is worsened in all areas but one when the Sahara is greened.

We thank the reviewer for the time s/he spent in revising and commenting our manuscript, and for the insightful comments, which helped in improving the paper. We agree with the reviewer that the Saharan greening doesn't improve the agreement with proxies. However, we highlight that the impact of the Sahara greening is larger on NAO in winter (see the Kolmogorov-Smirnov test), and WR frequencies show larger changes in the Green Sahara experiment, with respect to the PMIP experiment.

**Recommendation**

My only main comment is that some of the key findings could be more clearly articulated in the abstract and the conclusions. I also have a few minor suggestions about clarity that are listed below. Otherwise, this is a valuable study and I look forward to seeing it published.

We agree with the reviewer that the key findings should be better presented in the abstract and more deeply discussed in the conclusions. Please find in the following our responses to your comments.

**Main comments**

A key finding here is that the MH_PMIP simulation performs better than the MH_GS in all regions but one according to the Cohen Kappa scoring. I think this needs to be clearly stated in the abstract.

We agree with the reviewer that this aspect should be mentioned in the abstract. A line has been added:

"Although the prescription of vegetation in the Sahara does not improve the proxy-model agreement, this study provides…"

This slightly surprising finding could also benefit from further discussion - at the moment the Conclusions seem to argue for more realistic mid-Holocene simulations, but clearly there are nuances here. What could be the cause of this? One question that came up on reading was whether the highly idealised nature of the GS simulation setup could play a role?

We believe that the proxy-model disagreement originates in part from some local inconsistencies of proxy reconstructions, often showing ambiguous regional features that are difficult to reconcile with the large-scale circulation patterns simulated by global climate models. However, we agree with the reviewer that the highly idealised nature of our simulation setup may contribute to these discrepancies. The experimental design, initially conceived to improve the proxy-model agreement specifically within the Sahara, was extended to explore potential remote impacts both in the Tropics (see Pausata et al. 2017a, b) and at mid-latitudes (this manuscript). We now highlight this aspect more clearly in the discussion of the proxy-model agreement in Section 5:

"Furthermore, it should be noted that the simulation setup is highly idealised, initially tailored to enhance the representation of the MH precipitation in the Sahara and to improve the regional proxy-model agreement. While this approach has yielded insights into specific climate impacts, such as those associated

with Sahara greening, the broader applicability to global mid-Holocene climate scenarios is limited. The improvement of the global proxy-model agreement would benefit from more refined MH climate modelling strategies, such as prescribing more realistic vegetation across latitudes and considering the seasonal vegetation cycle, which could better account for the nuanced large- and local-scale climate feedbacks that are critical for understanding past climates (see e.g., Swann et al., 2014)".

**Minor Comments**

In the discussion of mid-latitude temperature change it would be worth referring to Bartlein et al (2017) who looked at this issue in multiple models.

We thank the Reviewer for suggesting this reference that we missed. The paper is now referred in the Introduction.

Figures: There are a lot of figures here and the reader has to jump between the main text and the appendix figures quite a lot. Could you consider moving one or more of these figures into the main text to reduce this. At least figure A2 would be better in the main text.

Fig. A2 is now moved to the main text, as new Fig. 4.

Lines 45-47: "However, the interpretation of these climatic changes, particularly on temperature and precipitation patterns, as indicated by proxies, seem potentially inconsistent with the suggested changes in the atmospheric circulation (e.g., the positive-to-negative shift in the NAO/AO phase)."

The sentence is rather obscure indeed, it is now rephrased:

"(e.g., a drier eastern North America, warmer Scandinavia and colder Mediterranean would be inconsistent with a positive-to-negative shift in the NAO/AO phase)".

Line 231: "However, the difference in the NAOI distributions between the MHGS and MHPMIP experiments is less significant (p<0.11)".

Do you mean not significant or just less?

We set the threshold for significance at p<0.05, however there is a signal of shifting NAOI distribution between the MHGS and MHPMIP experiments. We changed "less" to "weakly".

Lines 232-237: "Circulation and surface anomaly patterns associated with the NAO positive phase in the MHPMIP (not shown) and the MHGS experiments (Fig. 7c, d) are very similar.  …. In particular, the thermal and rainfall anomalies are more pronounced when Saharan greening is taken into account, due to the significant difference in the NAO phase shift with respect to the PI period."

These two statements seem contradictory to me. Please could you clarify?

Because of the shift of the NAO from a prevailing positive phase into prevailing negative, we expect to see temperature and precipitation anomalies associated with a negative NAO. Because we show that in MHGS the shift is more negative than in MHPMIP, those anomalies will be more pronounced in MHGS than in MHPMIP. However, because the differences in the NAO phase in MHGS and MHPMIP is only weakly significant, we replace "significant" with "larger".

Line 287: "… suggesting that the effect of the Saharan greening on the atmospheric circulation and the associated thermal and rainfall anomalies amplifies the changes driven solely by the orbital forcing"

Can you provide any potential explanations for this?

An analysis of the physical mechanisms responsible for the changes in the frequency of the atmospheric WRs in the different experiments is beyond the scope of this paper. However, in Section 5 we discuss a possible explanation of how enhanced deep convection in the Sahara adds on the orbital driven changes:

"The responses in temperature, precipitation and atmospheric dynamics are more pronounced in the MHGS simulation, indicating the significant influence of the Saharan greening on climate in the Northern Hemisphere. There is a broad literature on tropical-extratropical interactions triggered by tropical forcings such as the Indian monsoon and the El Niño/Southern Oscillation [e.g., Hoskins and Ambrizzi (1993); Rodwell and Hoskins (1996)]. More recently, the African monsoon has been indicated as a possible source of tropical-extratropical teleconnections (Gaetani et al., 2011; Nakanishi et al., 2021), reinforcing the hypothesis that the strengthening of deep convection in northern Africa associated with the Saharan greening could lead to climate impacts in the extratropics".

Lines 367-369: "This suggests that the MHGS simulation more effectively captures precipitation patterns compared to seasonal temperature patterns across the mid-latitudes"

Shouldn't this be MH_PMIP as the MH_GS runs has lower scores for every region except Asia?

This sentence refers to the comparison between model representation of precipitation and seasonal temperature anomalies in MHGS only, it is not a comparison with MHPMIP. A similar behaviour is also seen in MHPMIP.

Line 361: "numerous inconclusive MH proxy records"

Could you define what you mean by inconclusive?

We label as inconclusive those proxy records not providing a robust estimation of the change (or no change). See Section 2. A sentence has been added to the text:

"(as defined above, those record not providing a robust estimation of change or indicating no change)".

Line 405: "In addition, it is shown that the simulated Saharan greening drives"

Shouldn't this be the mid-Holocene orbit as the modes are similar in both GS and PMIP?

In Table 2 we show that, with respect to the MHPMP experiment, MHGS drives large changes in the frequency of occurrence of most of the WRs (NAO+, NAO-, SB in winter; NAO+, AR in summer). The spatial patterns do not change much among the three simulations, actually.

Line 409-410: "the changes driven by the Saharan greening in large-scale circulation indicate plausible explanations for the proxy evidence."

It's not clear what this really means but it sounds like its contradicting the Cohen's Kappa scores which show that the GS simulation is mostly worse than the PMIP simulation?

We agree that the Cohen's Kappa index shows improvements in only Asia in the MHGS simulation. However, we here highlight that regional inconsistencies in proxy reconstructions make difficult the link with large scale circulation patterns, with the exception of Asia, where proxy records show a more spatially homogeneous signal. Moreover, an overall more plausible association between precipitation

reconstructions and changes in the large-scale circulation is provided by the MHGS experiment. We rephrase for clarification:

"the changes driven by the Saharan greening in the large-scale circulation indicate plausible explanations for the proxy evidence, especially for precipitation".

Line 412-414: "Furthermore, this modelling exercise also highlights the need for more refined MH climate modelling, such as prescribing realistic vegetation across latitudes and considering the seasonal vegetation cycle, to account for large- and local-scale climate feedbacks."

These sound like sensible suggestions but they come slightly out of the blue here right at the end. Where do these ideas come from? Could you link them to your work or other studies in some way?

We rephrased this sentence in response to one of your main comments above:

"Furthermore, it should be noted that the simulation setup is highly idealised, initially tailored to enhance the representation of the MH precipitation in the Sahara and to improve the regional proxy-model agreement. While this approach has yielded insights into specific climate impacts, such as those associated with Sahara greening, the broader applicability to global mid-Holocene climate scenarios is limited. The improvement of the global proxy-model agreement would benefit from more refined MH climate modelling strategies, such as prescribing more realistic vegetation across latitudes and considering the seasonal vegetation cycle, which could better account for the nuanced large- and local-scale climate feedbacks that are critical for understanding past climates (see e.g., Swann et al., 2014)".

**Technical Corrections**

Hermann et al (2018) is missing from the reference list?

Added, thanks.

Figures 2 and 3: could you make the lines slightly less thick - as they are they overlap a bit too much.

Done.

Contributions: "QZ run the simulations" -> "QZ ran the simulations".

Corrected, thanks.